



# Year-round record of near-surface ozone and "O₃ enhancement events" (OEEs) at Dome A, East Antarctica

Minghu Ding[1,2,*], Biao Tian[1,*], Michael Ashley[3], Zhenxi Zhu[4], Lifan Wang[4], Shihai Yang[5], Chuanjin Li[2], Cunde Xiao[2,6], Dahe Qin[2]

[1]State Key Laboratory on Severe Weather, Chinese Academy of Meteorological Sciences, Beijing 100081, China
[2]State Key Laboratory of Cryospheric Science, Northwest Institute of Eco-Environment and Resources, Chinese Academy of Sciences, Lanzhou 730000, China
[3]Department of Astrophysics, University of New South Wales, Sydney 2052, Australia
[4]Purple Mountain Observatory, Chinese Academy of Sciences, Nanjing 210034, China
[5]Nanjing Institute of Astronomical Optics & Technology, Chinese Academy of Sciences, Nanjing 210042, China
[6]State Key Laboratory of Earth Surface Processes and Resource Ecology, Beijing Normal University, Beijing 100875, China
*These authors contributed equally to this work.

*Correspondence to*: Minghu Ding (dingminghu@foxmail.com)

**Abstract.** To evaluate the characteristics of near-surface O₃ over Dome A (Kunlun Station), which is located at the summit of the east Antarctic Ice Sheet, continuous observations were carried out in 2016. Together with observations from the Amundsen-Scott Station (South Pole) and Zhongshan Station, the seasonal and diurnal O₃ variabilities were investigated. The results showed different patterns between coastal and inland Antarctic areas that were characterized by high concentrations in cold seasons and at night. The annual mean values at the three stations were $29.19 \pm 7.52$ ppb, $29.94 \pm 4.97$ ppb and $24.06 \pm 5.79$ ppb. Then, specific atmospheric processes, including synoptic-scale air mass transport, were analysed by Hybrid Single-Particle Lagrangian Integrated Trajectory (HYSPLIT) back-trajectory analysis and the potential source contribution function (PSCF) model. Long-range transport was found to account for the O₃ enhancement events (OEEs) during summer at Dome A, rather than efficient local production (consistent with previous studies in inland Antarctica). In addition, we observed OEEs during the polar night in the Dome A region, which was not previously found in Antarctica. To explain this unique finding, the occurrence of stratospheric intrusion (stratosphere-to-troposphere, STT) events was studied with the Stratosphere-to-Troposphere Exchange Flux (STEFLUX) tool. This finding suggested that STT events occurred frequently over Dome A and could account for 55% of the total polar night period. The occurrence probability of OEEs agreed well with STT events, indicating that the



STT process was the dominant factor affecting the near-surface $O_3$ over Dome A in the absence of photochemical reaction sources during polar night. This work provides unique information on ozone variation at Dome A and expands our knowledge regarding such events in Antarctica.

**Key words**: near-surface $O_3$; Antarctica; OEE; STT;

# 1 Introduction

Ozone ($O_3$) is a natural atmospheric component that is found both in the stratosphere and troposphere and plays a major role in the atmospheric environment through radiative and chemical processes. Ozone does not have direct natural sources (not emitted from the ground or vegetation) but rather is produced in the atmosphere, and its concentration ranges from a few ppb near the Earth's surface to approximately a few ppm in the stratosphere. Stratospheric $O_3$, which is produced as a result of the photolysis of molecular oxygen, forms a protective layer against the UV radiation from the Sun. By contrast, throughout the troposphere and at the surface, $O_3$ is considered a secondary short-lived air pollutant (Monks et al., 2015), and ozone itself is a greenhouse gas, such that a reduction in concentration will also have a direct influence on radiative forcing (Mickley et al., 2001; IPCC, 2013; Stevenson et al, 1998).

Ozone is produced in the troposphere through the oxidation of precursors such as CO or hydrocarbons in the presence of high concentrations of $NO_x$. As these precursors are localized and their lifetimes are generally short, the distribution of near-surface ozone, which is produced from anthropogenic precursors, is also localized and time-variant. In the presence of strong solar radiation ($\lambda$ < 424 nm), volatile organic compounds (VOCs) and $NO_x$ ($NO + NO_2$), ozone is photochemically produced and can accumulate to a hazardous level during favourable meteorological conditions (Davidson, 1993; Wakamatsu et al., 1996). In the case of $NO_x$-rich air, $NO_2$ is produced and accumulates via the reaction between NO and $HO_2$ or $RO_2$ (peroxy radicals), which is followed by the accumulation of $O_3$. However, in the case of $NO_x$-poor air, these proxy radicals react with ozone and lead to ozone loss (Lin et al., 1988). Experiments conducted in Michigan (Honrath et al., 2000a) and Antarctica (Jones et al., 2000) found that $NO_x$ can be produced in surface snow. This production





appears to be directly driven by incident radiation and photolysis of nitrate deposited in the snow (Honrath et al., 2000a, b).

Previous investigations suggested that $O_3$ production in the Antarctic planetary boundary layer (PBL) can be affected by several global/regional climate-related variables, i.e., changes in UV fluxes due to total $O_3$ variability over Antarctica (e.g., Jones and Wolff, 2003; Frey et al., 2015), variability in long-range air mass transport patterns (e.g., Legrand et al., 2016), and the depth of the continental mixing layers. Summer episodes of "$O_3$ enhancement events" (OEEs) have been observed in the Antarctic interior (e.g., Crawford et al., 2001; Legrand et al., 2009; Cristofanelli et al., 2018) and at coastal sites influenced by the air mass transport from the interior of the continent (e.g., Cristofanelli et al., 2011). This phenomenon was attributed to the photodenitrification of the summer snowpack, which can result in $NO_x$ emissions to the atmosphere and subsequent photochemical $O_3$ production (e.g., Jones et al., 1999, 2000). These processes are capable of driving the seasonality of near-surface $O_3$ over the Antarctic Plateau (e.g., Crawford et al., 2001; Legrand et al., 2009), thus potentially providing a significant input of $O_3$ to the whole Antarctic region (e.g., Legrand et al., 2016). Indeed, as shown in Cristofanelli et al. (2008) and Legrand et al. (2016), due to air mass transport, the photochemically produced $O_3$ in the PBL over the Antarctic Plateau can affect the $O_3$ variability thousands of km away from the emission area.

Moreover, the near-surface ozone concentrations at high-elevation sites can also be increased by the downward transport of ozone-rich air from the stratosphere during deep convection and stratosphere-to-troposphere transport (STT) events (e.g., Bonasoni et al., 2000; Stohl et al., 2000; Lefohn et al., 2012; Jia et al., 2015; Ma et al., 2014; Langford et al., 2009, 2015; Lin et al., 2012a, 2015a; Yin et al., 2017). The earliest study carried out by aircraft flight NSFC-130 over the Ellsworth Mountains of Antarctica in 1978 found that mountainous terrain may induce atmospheric waves that propagate across the tropopause. The tropospheric and stratospheric air may be mixed, leading to an increase in the tropospheric ozone concentration (e.g., Robinson et al., 1983). Radio soundings at the Resolute and Amundsen-Scott Stations also showed the existence of transport from the stratosphere to the troposphere, and the flux could reach up to $5*1010$ mol/cm$^2$/s (e.g., Gruzdev, 1993). Recently,



Traversi et al. (2014, 2017) suggested that the variability of air mass transport from the stratosphere to the Antarctic Plateau could affect the nitrate content in the lower troposphere and the snowpack.

85    Currently, the climatology of tropospheric ozone over Antarctica is relatively understudied because observations of year-round near-surface ozone have been tied to manned research stations. These stations are generally located in coastal Antarctica, except for the South Pole (SP) and Concordia continental stations on the East Antarctic Plateau. Thus, the only information currently available for the vast region between the coast and plateau are spot measurements of boundary layer ozone during

90    summer from scientific traverses (e.g., Frey et al., 2005) or airborne campaigns (e.g., Slusher et al., 2010). Moreover, the vertical profile of ozone in the troposphere cannot be measured by satellites because the high density of ozone in the stratosphere leads to the overestimation of tropospheric ozone by limb-viewing sensors. Estimates of total ozone in the tropical troposphere have been made by subtracting the stratospheric ozone column (determined by a limb-viewing sensor) from the total ozone

95    column (measured by a nadir-viewing sensor) (Fishman et al., 1990). In other words, tropospheric profiles cannot be obtained by satellites, and we cannot examine the spatial distribution of near-surface ozone from space. As a result of these limitations, a dearth of information exists regarding the spatial gradient of near-surface ozone across Antarctica and how it varies throughout the year.

    To better understand the spatial variations and the source-sink mechanisms of near-surface ozone

100   in Antarctica, near-surface ozone concentrations were monitored during 2016 at Dome A (DA) and the Zhongshan Station (ZS). Together with records from the Amundsen-Scott Station (SP), we analysed specific processes that affect the intra-annual variability in surface $O_3$ over the East Antarctic Plateau; in particular, we determined (i) the synoptic-scale air mass transport within the Antarctic interior and (ii) STT transport. This study broadens the understanding of the spatial temporal variations in the

105   near-surface ozone concentration and transport processes that impact tropospheric ozone over high plateaus.



## 2 Sites and methods description

### 2.1 Sites and instruments

The Kunlun Station (80°25'02"S, 77°06'59"E, altitude 4087 m) is located in the DA area, which is the summit of the east Antarctic Ice Sheet (Figure 1). On the 1st of Jan 2016, we deployed a 2B-205 UV absorptive near-surface ozone analyser during the 33rd Chinese National Antarctic Research Expedition. Because of the extreme environment, we could only calibrate the instrument during austral summer with a Model-49iPS UV absorptive ozone calibrator. However, the related coefficients (r) were all greater than 0.99 in Jan 2016 and Jan 2017.

The ZS (69°22'12"S, 76°21'49"E, altitude 18.5 m) is located at the edge of the east Antarctic Ice Sheet (Figure 1), where we installed a UV absorption near-surface ozone analyser (EC9810A) for long-term near-surface ozone monitoring. The observational frequency was 3 min, and the data were transported in real time to Beijing. Furthermore, to prevent data losses, a CR1000 data logger was used to record the data output in real time. Every three months, the ozone analyser was calibrated using the EC9811 ozone calibrator, and 5 standard concentrations of ozone gas were generated for each calibration. The calibration concentration and measured concentration underwent correlation analysis, and seasonal calibration results were generated every three months. In 2016, 5 calibrations were made, and the related coefficients (r) were all greater than 0.9995.

The Amundsen-Scott Station (89°59'51.19 "S, 139°16'22.41" E, altitude 2835 m) is located at the SP and operated by the United States. The near-surface ozone data were downloaded from the World Data Centre for Greenhouse Gases under the Global Atmosphere Watch programme.

Surface ozone concentration data were processed to obtain the hourly mean data. The variance test was used to remove abnormal data values based on the formula $| x_i - x | > 3\sigma$, where $x_i$ is the measured value, x is the time series mean and $\sigma$ is the standard deviation. After processing, 99.2%, 91.6%, and 99.5% of the hourly mean data were retained from the Amundsen-Scott Station, Kunlun Station and ZS, respectively.



## 2.2 Meteorological simulations

The Hybrid Single-Particle Lagrangian Integrated Trajectory (HYSPLIT) backward air mass trajectory model was previously applied to atmospheric research in Antarctica (Legrand et al., 2009; K. Hara et al., 2011). To analyse the impact of variable air mass sources and the intrusion of stratospheric ozone, we also used the HYSPLIT model in this paper. Backward trajectories and clusters were calculated using the US National Oceanic and Atmospheric Administration (NOAA)-HYSPLIT model (Draxler and Rolph, 2003; http://ready.arl.noaa.gov/HYSPLIT.php), which is a free software plug-in for MeteoInfo (Wang, 2014; http://meteothink.org/). Gridded meteorological data for backward trajectories in HYSPLIT were obtained from the Global Data Assimilation System (GDAS-1) operated by NOAA with 1°*1° latitude and longitude horizontal resolution (http://www.arl.noaa.gov/gdas1.php). The initial calculation height was set to 20 m, and the back-up time was 120 h, as suggested by Legrand et al. (2009). The ECMWF ERA-Interim data (Dee et al., 2011) were used to analyse the upper-troposphere and lower-stratosphere structures of the meridional cross section over DA.

## 2.3 Potential source contribution function

As in Yin et al. (2017), the potential source contribution function (PSCF) assumes that back trajectories arriving at times of high mixing ratios likely point to significant pollution directions (Ashbaugh et al., 1985). This function was often applied to locate air masses associated with high levels of near-surface ozone at different sites (Kaiser et al., 2007; Dimitriou and Kassomenos, 2015). In this study, the PSCF was calculated using HYSPLIT trajectories. The top of the model was set to 10000 m a.s.l. The PSCF values for the grid cells in the study domain were calculated by counting the trajectory segment endpoints that terminated within each cell (Ashbaugh et al., 1985). The number of endpoints that fell in the ijth cell was designated $n_{ij}$. The number of endpoints in the same cell with arrival times at the sampling site that corresponded to PM concentrations that were higher than an arbitrarily set criterion was defined as $m_{ij}$. The PSCF value for the ijth cell can then be defined as

$$\text{PSCF}_{ij} = \frac{m_{ij}}{n_{ij}} \qquad (1)$$



The PSCF value can be interpreted as the conditional probability. The concentrations of a given analyte greater than the criterion level are related to the passage of air parcels through the $ij$th cell during transport to the receptor site. That is, cells with high PSCF values are associated with the arrival of air

parcels at the receptor site, which has near-surface ozone concentrations that are higher than the criterion value. These cells are indicative of areas with 'high potential' contributions of the constituent. Identical PSCF$_{ij}$ values can be obtained from cells with very different counts of back-trajectory points (e.g., grid cell A with $m_{ij}$ = 5000 and $n_{ij}$ = 10000 and grid cell B with $m_{ij}$ = 5 and $n_{ij}$ = 10). In this extreme situation, grid cell A has 1000 times more air parcels passing through it than grid cell B.

Because the particle count in grid cell B is sparse, the PSCF values in this cell are highly uncertain. To account for the uncertainty due to the low values of $n_{ij}$, the PSCF values were scaled by a weighting function $W_{ij}$ (Polissar et al., 1999). The weighting function reduced the PSCF values when the total number of endpoints in a cell was less than approximately 3 times the average number of end points per cell. In this case, $W_{ij}$ was set as follows:

$$W_{ij} = \begin{cases} 1.00 & n_{ij} > 3Nave \\ 0.70 & 3Nave > n_{ij} > 1.5Nave \\ 0.42 & 1.5Nave > n_{ij} > Nave \\ 0.05 & Nave > n_{ij}, \end{cases} \quad (2)$$

where Nave represents the mean $n_{ij}$ of all grid cells. The weighted PSCF values were obtained by multiplying the original PSCF values by the weighting factor.

## 3 Near-surface O₃ variability

### 3.1 Mean concentration

The annual mean concentrations of near-surface ozone at DA, the SP, and the ZS were 29.19 ± 7.52 ppb, 29.94 ± 4.97 ppb and 24.06 ± 5.79 ppb, respectively. The maximum concentration reached 42.50 ppb, 46.44 ppb and 32.81 ppb, and the minimum concentrations were 14.00 ppb, 10.89 ppb and 9.88 ppb, respectively. The inland stations experienced higher near-surface ozone concentrations than the coastal station.





There were also obvious differences between polar day and polar night at all stations. In Figure 2, we used different shading colours to signify the polar day and polar night. The average concentrations of near-surface ozone during polar night at DA, the SP and the ZS were 34.12 ± 4.31 ppb, 31.52 ± 3.92 ppb and 28.72 ± 1.28 ppb, respectively, and much lower concentrations appeared during non-polar night, with corresponding values of 26.13 ± 6.95 ppb, 28.14 ± 5.84 ppb and 23.14± 5.89 ppb, respectively.

Interestingly, the SP had the highest near-surface ozone concentration during polar day/non-polar night, whereas at DA, the highest concentration occurred during polar night and the largest variation occurred at this site.

## 3.2 Seasonal variation

    In this part, we define Oct-Mar as the warm season and Apr-Sept as the cold season, similar to the

definition of polar day and night.

    In agreement with previous studies (Oltmans et al., 1976; Gruzdev et al., 1993; Ghude S D et al., 2010), the concentrations of near-surface ozone at the three stations were high and stable during the cold season and low and variable during the warm season (Figure 3). In Antarctica, the emissions of ozone precursors are generally less than those at mid and low latitudes, whereas ultraviolet radiation is

relatively strong; thus, the depletion effect is much greater than the effects from photochemical reactions during the warm season. The lowest monthly concentrations of near-surface ozone at DA, the SP and the ZS appeared in February (17.34 ppb), February (22.45 ppb) and January (15.29 ppb), respectively. The monthly fluctuations in near-surface ozone at the ZS was generally small compared to those at the other two stations due to short polar nights; however, this station showed large seasonal

variations because of the strong ultraviolet radiation in low latitude areas and the presence of ozone depletion events controlled by coastal bromine elements (Wang et al., 2011; Prados-Roman et al., 2017). Figures 2 and 3 show that the near-surface ozone showed obviously larger variations at DA than at the SP during the polar night. As air masses from low-mid latitudes do not often reach the DA area where the near-surface ozone is high due to the existence of a strong polar vortex system in winter (Schoeberl

et al., 1992; van den Broeke et al., 2002), this phenomenon seems abnormal and implies an uncovered ozone source for the DA area.



Specifically, the largest standard deviation was observed in October at DA because of multiple influences, including photochemical reactions by ozone precursors and ultraviolet radiation, photolysis reactions by strengthened ultraviolet radiation, and external air masses from the coast. However, at the Amundsen-Scott Station, the largest standard deviation was observed in December, similar to the characteristics at DC from November to December (Legrand et al., 2009; Cristofanelli et al., 2018).

As mentioned in the introduction section, mountainous topography/mountain waves may disturb advection transport in the stratosphere and lead to downward transportation to the troposphere (Robinson et al., 1983). DA is the summit of the east Antarctic Ice Sheet, and the tropospheric depth is only ~4.6 km (Fu et al., 2015), which favours exchange between the stratosphere and troposphere. However, the topography in this area is very flat and creates a disadvantage for mountain waves. Is there ozone exchange happening? We will analyse and discuss this question in section 4.

### 3.3 Diurnal variation

To characterize the typical monthly $O_3$ diurnal variations at these three stations, we analysed the average diurnal variation characteristics during a polar day, polar night and normal day. At DA, the average diurnal concentrations at polar night and on a normal day were relatively steady, with values of 34.12±0.15 ppb and 27.51±0.25 ppb. During the polar day, the average diurnal concentration fluctuated greatly (25.1±0.74 ppb). At the SP, the average diurnal variations during the polar day and polar night were stable, with values of 28.15±0.15 ppb and 31.51±0.04 ppb. Because of the limited number of normal days, the diurnal concentration fluctuated with an average of 30.06±0.91 ppb. At the ZS, the average diurnal concentrations at polar night and on a normal day varied relatively steadily, with values of 28.71±0.15 ppb and 24.56±0.25 ppb. The average diurnal variations in different time periods were not obvious, and the average diurnal concentrations of the three stations fluctuated within a range of less than 1 ppb, indicating that daily photochemistry reactions did not have the main impacts on the overall characteristics of near-surface ozone at the three stations. The magnitude of the diurnal variation was low, which is similar to the variations found at other Antarctic stations (Gruzdev et al., 1993; Ghude S D et al., 2010; Oltmans et al., 1976). This finding suggests that synoptic transport somehow controls the





overall O$_3$ variability even during the photochemical "active" summer period, as found by Neff et al. (2008).

During non-polar night, the concentration of near-surface ozone at the SP was approximately 3 ppb higher than that at DA, and the concentrations at the SP are similar to those at the Concordia Station (e.g., Legrand et al., 2016; Cristofanelli et al., 2018). During the polar night, the concentration of near-surface ozone at DA was approximately 3 ppb higher than that at the SP. In contrast to the SP and ZS, the standard deviation of the diurnal near-surface ozone concentration at DA was the largest

(mostly >0.7 ppb), and the near-surface ozone declined at a rapid rate (0.25 ppb/h from UTC 1:00 to UTC 9:00). Generally, NOx/hydrocarbons are considered the most important precursors of O$_3$ and are produced in surface snow under strong solar radiation in Antarctica (HNO$_3$ + hυ →OH + NO$_2$ (λ≤ 604 nm); Galbally and Allison, 1972). However, the process of O$_3$ production (OH+NO$_2$* →HONO + O$_3$ (λ≤ 393 nm); Atkinson R,et al., 1997) mainly acts under UV photolysis conditions at 253 K and 308

nm (Honrath et al., 2000a, b; Zhu et al., 2012). At the SP and ZS, this chain reaction produced near-surface ozone when photochemical losses occurred, whereas at DA, the surface snow temperature remained lower than 233 K (Campbell et al., 2013; Ding et al., 2015), which would restrain the photolysis of nitrate. This finding implies that an extremely cold environment is the main reason for the large variations in diurnal near-surface ozone at DA.

**4 Ozone under OEEs at the Kunlun Station during polar night**

**4.1 Identification of OEEs**

The algorithm proposed in Cristofanelli et al. (2018) was applied in this paper to analyse OEEs at the three stations. The algorithm includes two steps: the first step is to generate a sinusoidal curve to fit the annual cycle of the daily mean O$_3$ value, which represents an "undisturbed" O$_3$ annual cycle (not

affected by the occurrence of OEEs). In the second step, we need to calculate the probability density function (PDF) of the deviations from the sinusoidal fit, considering all the daily data. Figure 5 a, 5b, 5c shows the OEEs and "NO O$_3$ enhancement events" (NOEEs) at these three stations, while Figure 5d, 5e, 5f reports the PDF of the calculated deviations. Then, we applied a Gaussian fit to the obtained PDF. As



reported by Giostra et al. (2011), the Gaussian distribution corresponds to a well-mixed state, given the
hypothesis that instrumental errors and natural background variability follow a Gaussian distribution.
The deviations from the Gaussian distribution (calculated by the Origin 9© statistical tool) would be a
sign of observations affected by abnormal variability. To obtain a threshold value for selecting
non-background $O_3$ daily values possibly affected by "anomalous" $O_3$ enhancements, a further Gaussian
fit was calculated for the PDF points falling above 1 σ (standard deviation) of the Gaussian PDF. The
intersection between the two curves was considered our threshold value (i.e., 3.4 ppb at the SP, 3.4 ppb
at DA, 2.5 ppb at the ZS).

In total, 42 days at DA were found to be affected by anomalous OEEs: 14.29% in January, 2.38%
in May, 14.29% in Jun, 4.76% in July, 11.90% in August, 4.76% in November and 47.62% in
December (Figure 5e, blue bars). This result clearly indicates that half of the anomalous days occurred
in December, followed by January and June. At the SP, 36 days with OEEs were found in 2016: 44.4%
in January, 30.6% in November, and 25% in December (Figure 5d, grey bars). Apparently, OEEs occur
only in summertime at the SP. There were more days with OEEs at the ZS: 53 days with the highest
fraction in April (33.96%), followed by September (18.87%), January (13.21%), October (11.32%),
November (11.32%), December (5.66%) March (3.77%) and May (1.89%) (Figure 5f, yellow bars).

From the results above, the SP was characterized by concentrated OEE occurrences, and the ZS
had the most scattered OEEs pattern. In addition, all OEEs at the SP and ZS occurred during the
Antarctic warm season, and no OEEs occurred during the polar night, which is similar to the pattern in
DC (Cristofanelli et al., 2018). In contrast, the OEEs also occurred during the polar night in DA, and the
number of OEE occurrence days accounted for up to 33% of the total number of events throughout the
year. Previous studies (e.g., Legrand et al., 2016; Cristofanelli et al., 2018) carried out in DC showed
that the $O_3$ variability at DC could be associated with processes occurring at long temporal scales. In
addition, the accumulation of photochemically produced $O_3$ during transport of air masses was the main
reason for OEEs, whereas the stratospheric intrusion events had only a minor influence (up to 3%) on
OEEs. This finding cannot explain the temporal occurrence pattern of OEEs at DA. To determine the
unknown cause, we investigated the synoptic-scale air mass transport and the STT processes during the
events.



## 4.2 Role of synoptic-scale air mass transport

The observation of a secondary maximum of ozone in November–December at the inland Antarctic sites was first reported for the SP by Crawford et al. (2001) and was attributed to

photochemical production induced by high NOx levels in the atmospheric surface layer, which were generated by the photodenitrification of the Antarctic snowpack (same as Davis et al., 2001). At DC, a secondary maximum in November–December 2007 was also reported by Legrand et al. (2009), proving that photochemical production of ozone in the summer takes place over a large part of the Antarctic Plateau. A further study by Legrand et al. (2016) found that the highest near-surface $O_3$ summer values

were observed within air masses that spent extensive time over the highest part of the Antarctic Plateau before arriving at DC. To investigate the possible influence of synoptic-scale air mass circulation on the occurrence of OEEs at DA, 5-day HYSPLIT back trajectories were applied (Figure 6). In particular, we used the PSCF to calculate the conditional probabilities and identify the geographical regions related to the occurrence of OEEs at DA (e.g., Hopke et al., 1995; Brattich et al., 2017). The residence time over

each i, j grid cell of the back-trajectory spatial domain (discretized into square grid cells of 0.5° × 0.5°) was calculated by selecting the number of trajectory endpoints falling within grid cell i, j over the whole measurement period (nij). Then, in the same way, the residence time over each i, j grid cell corresponding to a measurement period under an OEE was calculated (mij). The PSCF for grid cell i, j was then computed according to Brattich et al. (2017):

$$\mathrm{PSCF} = \frac{m_{ij}}{n_{ij}} = \frac{residence\ time\ of\ the\ air\ parcel\ during\ OEEs}{residence\ time\ of\ the\ air\ parcel} \qquad (3)$$

The results are shown in Figure 6, which clearly highlights the connection between the occurrence of OEEs at DA and air masses travelling over the eastern Antarctic Plateau. The PSCF values ranged from 40% to 80% in the DA region, while air masses advected from the East Antarctic Plateau were characterized by high PSCF values. The aboveground level (A.G.L. m) and pressure altitude (hPa) of

the back-trajectory points during OEEs were also simulated to identify the synoptic-scale air mass transport. The results showed that 66% of the OEE trajectories were lower than 200 A.G.L. m (Figure 6a and 6b), and all pressure heights were between 500 hPa and 750 hPa (Figure 6d and 6e), which indicated that most of the backward air mass trajectories affecting the OEEs over DA travelled along



the channel of the boundary layer over the East Antarctic Plateau. During non-polar night, the emissions

from snowpack could be a source of $O_3$ (e.g., Jones et al., 1999, 2000); hence, the concentration of $O_3$ in
the air masses may accumulate and lead to OEEs at DA.

Interestingly, the $O_3$ at DA was obviously higher than that at the other inland stations during polar
night, when photochemical reactions do not occur. In addition to the long distance to the coast, the high
altitude and the long-term sustainable polar vortex in the DA region, the horizontal transportation from

marine, sea ice and coastal areas had little impact on the near-surface ozone variation at DA. This
finding indicates that air mass transportation from the East Antarctic Plateau was not the main reason
for OEEs at DA during polar night. Additionally, 12.5% of the backward air mass trajectories came
from the pressure altitude below 550 hPa in Figure 6, which may be accompanied by STT events and
lead to downward transport of stratospheric high-concentration ozone . We will continue this discussion

in the next section.

## 4.3 Role of STT events

### 4.3.1 Identification of STT events

Several methods can be applied to study STT events. One method is the chemistry-climate hindcast
model GFDL-AM3, which Lin et al. (2017) used to evaluate the increasing anthropogenic emissions in

Asia, and Xu et al. (2018) used to examine the impact of direct tropospheric ozone transport at the
Waliguan Station. Stratosphere-to-Troposphere Exchange Flux (STEFLUX) is a novel tool to quickly
obtain reliable identification of SI events occurring at a specific location and during a specified time
window, and the tool was recently developed by Putero et al. (2016). STEFLUX relies on a compiled
stratosphere-to-troposphere exchange climatology, which makes use of the ERA-Interim reanalysis

dataset from the ECMWF, as well as a refined version of a well-established Lagrangian methodology.
STEFLUX is able to detect stratosphere intrusion events on a regional scale, and it has the advantage of
retaining additional information concerning the pathway of stratospheric-affected air masses, such as
the location of tropopause crossing and other meteorological parameters along the trajectories. All the
studies did so by defining a potential vorticity value for the tropopause. Here, we use STEFLUX to

identify STT events and define the height of tropospheric potential vorticity PVU = 2. The flux of


transport from the stratosphere to the troposphere is positive if PVU > 2 at 300 hPa (some studies defined 1.5 PVU as the threshold) (Vaughan et al., 2010; Cox et al., 2010). From a large set of global trajectories available each day, the trajectories that cross the tropopause within the first 24 h are retained (see Škerlak et al., 2014 for more details), and this period is determined as one STT event.

Drawing on the algorithm suggested in previous studies (e.g., Brattich et al., 2017), the potential vorticity and ozone mixing ratio of different pressure altitudes (hPa) at 0:00, 6:00, 12:00, and 18:00 every day were analysed during polar night (May, Jun, July, August) at DA. If the potential vorticity of 550 hPa height over DA was greater than 2 PVU, it was considered an STT event. Together with meteorological observations and ground ozone concentration data, the main causes of the outbreaks of
OEEs during polar night at DA were analysed.

## 4.3.2 Role of STT events at DA

As mentioned above, the polar night at DA lasts from April 15 to August 27. All of the backward air mass trajectories during OEEs come from the East Antarctic Plateau, partly due to its high altitude
and long distance to the coast. During polar night, the photochemical reactions on the snow surface and in the atmosphere come to a halt. In the absence of local sources, the occurrence of OEEs during polar night at DA can arise from only the downward transmission of high-concentration ozone in the stratosphere.

The ozone mixing ratios and potential vorticity over DA are shown in Figure 7b and 7c. The
temporal variability of the ozone vertical profile is similar to the potential vorticity. The dashed red line in Figure 7c, which represents the tropospheric top (2 PVU), frequently moved to near the ground during the study period. With every appearance of a deepened troposphere, the ozone mixing ratio increased accordingly. In detail, it could be found that (1) from July to September, extensive turbulence occurred below 300 hPa atmospheric potential vorticity, and each large disturbance corresponded to an
obvious increase in near-surface ozone. (2) The high ozone concentration was distributed at barometric altitudes between 50 hPa and 200 hPa, and its fluctuations agreed with the variations in potential vorticity. This result indicated that the turbulence occurring near the tropopause could directly affect the





ozone concentration. (3) Every rapid increase in potential vorticity was accompanied by rapid cooling in the boundary layer (Figure 7d).

To quantitatively analyse the influence of STT events on OEEs, we examined the appearance of STT events above DA and found that STT events (550 hPa PV>2 PVU) accounted for 55% of the polar night in 2016. Furthermore, we calculated the probability of occurrence of events-A (NOEE (A0), OEE (A1)) and events-B (NSTT event (B0), STT event (B1)) at DA during polar night (Table 1). The probability of the simultaneous occurrence of two events was 55.21%, and the probability of occurrence

of event-A and non-occurrence of event-B was only 16.29%.

In addition, the average ozone growth rate reached 0.29 ppb/h during OEEs, while the average ozone growth rate was -0.06 ppb/h during NOEEs (Figure 8). The statistical scatter distribution showed that 97% of OEEs occurred when the wind speed was less than 4 m/s, and 70% of OEEs occurred when the potential vorticity of 550 hPa was greater than 2 PVU. In total, 57% of OEEs were accompanied by

high potential vorticity and low wind speed. This indicates that OEEs prefer lower PBLs and weak winds, which favours the occurrence of STT appearance.

In other words, STT events occur frequently during the polar night at DA, and the transmission of high concentrations of ozone from the stratosphere is the main reason for the abnormal increase in near-surface ozone during the polar night at DA.

4.3.3 Analysis of a typical OEE

To better illustrate the difference between DA and the SP (Figure 7a), we take an OEE event on July 25th as an example and carried out HYSPLIT analysis. A total of 10 trajectories and their 3D profiles were generated hourly from 00:00 to 10:00 (Figure 9). All air masses were transported along the same route: the air mass quickly climbed to the ground level of AGL700 m (~4700 m a.s.l.) from

July 21st to 22nd, and then the air mass quickly sank below 200 m above the ground and transported to DA. At the same time, the horizontal distribution of ozone concentration and potential vorticity at 550 hPa from July 21 to 29, 2016 (Figure 10 a, b) showed that within the range of 60°S ~ 90°S, the potential vortex corresponds well to the ozone concentration distribution, in other words, the high concentration ozone invading the air in the stratosphere corresponds to the high vortex area. In addition, the high





vortex region in figure 10 (a) corresponds to the dark region (low value region) in the water vapor image in figure 10 (c). The comparison of horizontal distributions of relative humidity and wind field at the height of 550 hPa, illustrated a jet stream region within the DA range. This was in consistent with the high ozone region in figure 10b. From the figure, it could be seen that the relative humidity in the jet stream was less than 40% (white area). These characteristics further indicated that it is a stratospheric

invasion event affected by jet stream significantly. To sum up, during this invasion, DA located in the entrance area of the jet stream and in the area of negative vorticity advection.

From diagnosis view, it can be seen that on the 25th, the potential vorticity of 550 hPa over DA was obviously >2 PVU (Figure 10a), which suggests that the edge of the stratospheric invasion region reached the ground of DA (the air pressure at DA is normally ~575 hPa). Obviously, the OEE incident

on July 25th was caused by an STT event.

## 5 Summary

Based on the in situ monitoring data during 2016 at DA, the variation, formation and decay mechanisms of near-surface ozone were studied and compared with those at the SP and ZS. The annual

mean concentrations of near-surface ozone at DA, the SP and the ZS were $29.19 \pm 7.52$ ppb, $29.94 \pm 4.97$ ppb and $24.06 \pm 5.79$ ppb, respectively. The near-surface ozone concentrations were obviously higher in winter/polar night with small fluctuations than in the other seasons, which is different from the patterns observed at low latitudes. The $O_3$ in inland areas was also higher than that over coastal land.

The diurnal variations showed nonsignificant regular patterns, and the range of the average diurnal

concentration fluctuation was less than 2 ppb at all three stations. The extremely low snow surface temperature in the DA area inhibited the photodecomposition of nitrate and led to weak $NO_x$ production, which could not favour $O_3$ formation during the daytime. Consequently, the diurnal variation at DA was larger than that at the other two stations and stations at low altitudes. During polar night, the average daily concentration of DA was significantly higher than that at the other two stations. These findings

also suggest that the synoptic transport somehow controls the overall $O_3$ variability, which has also been shown at the Halley Station, McMurdo Station, etc. (Neff et al., 2008; Frey et al., 2015).

The most important OEEs were first found during the polar night in the DA region, which has never been observed in previous studies. Statistical analysis of STT events and OEEs revealed that the transportation from the stratosphere to the troposphere should be the dominant reason for this

phenomenon.

The characteristics and mechanisms of near-surface ozone revealed in this paper have important implications for better understanding the formation and decay processes of near-surface ozone in Antarctica, especially over the plateau areas. Nevertheless, the lack of observations restricted our ability amass more information. In addition, the Lagrangian tool based on ERA-Interim data may not have high

enough spatial and vertical resolution to resolve the complex transport of stratospheric air masses down to the Antarctic Plateau during STT events (Mihalikova and Kirkwood, 2013). Long-term sustained observations at (but not only at) Dome A, Dome C, Dome F, the SP and Vostok would greatly help in the future. Atmospheric chemical models equipped with polar processes are also valuable.

## 6 Data availability

All the data presented in the paper are available for scientific purposes upon request to the corresponding author (Minghu Ding, dingminghu@foxmail.com).

**Author contribution**

Minghu Ding and Biao Tian designed experiments and wrote the manuscript; Minghu Ding carried out experiments; Biao Tian analyzed experimental results. Michael Ashley, Zhenxi Zhu, Lifan Wang, Shihai Yang, Jie Tang, Chuanjin Li, Cunde

Xiao and Dahe Qin discussed the results.

**Acknowledgements:**

This work is financially supported by the National Natural Science Foundation of China (41771064), the Strategic Priority Research Program of Chinese Academy of Sciences (XDA20100300) and the Basic Fund of the Chinese Academy of Meteorological Sciences (2018Z001). The observations were carried out by during the Chinese National Antarctic Research





Expedition at the Zhongshan Station and the Kunlun Station. We are also grateful to NOAA for providing the HYSPLIT model and GFS meteorological files. Yaqiang Wang is the developer of MeteoInfo and provided generous help for the paper.

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







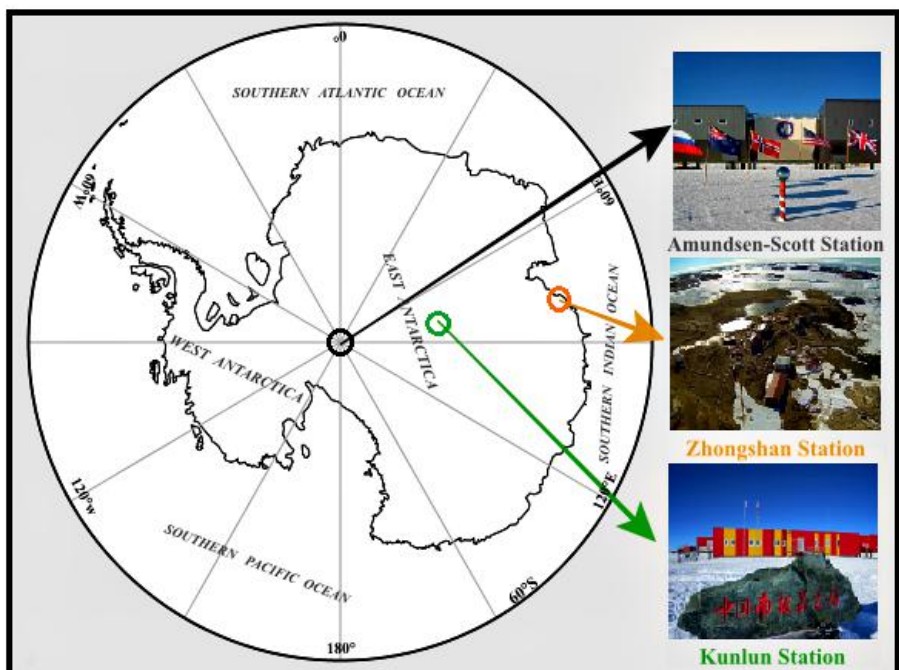

**Figure 1. Amundsen-Scott Station (South Pole), Kunlun Station (Dome A) and Zhongshan Station in Antarctica.**





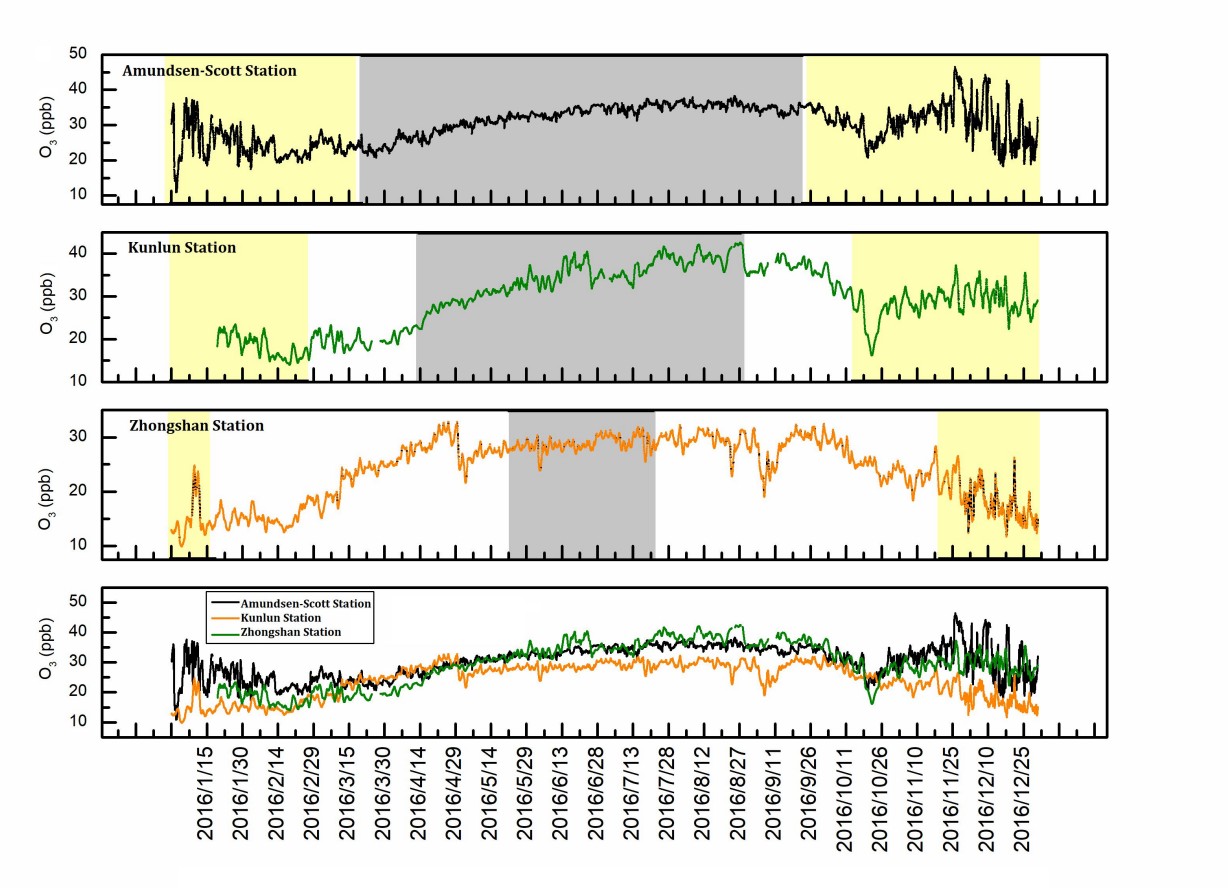

**Figure. 2. Time series of near-surface ozone at the Amundsen-Scott Station, Kunlun Station and Zhongshan Station**
655             **during 2016. Yellow shading signifies polar day and grey shading signifies polar night.**





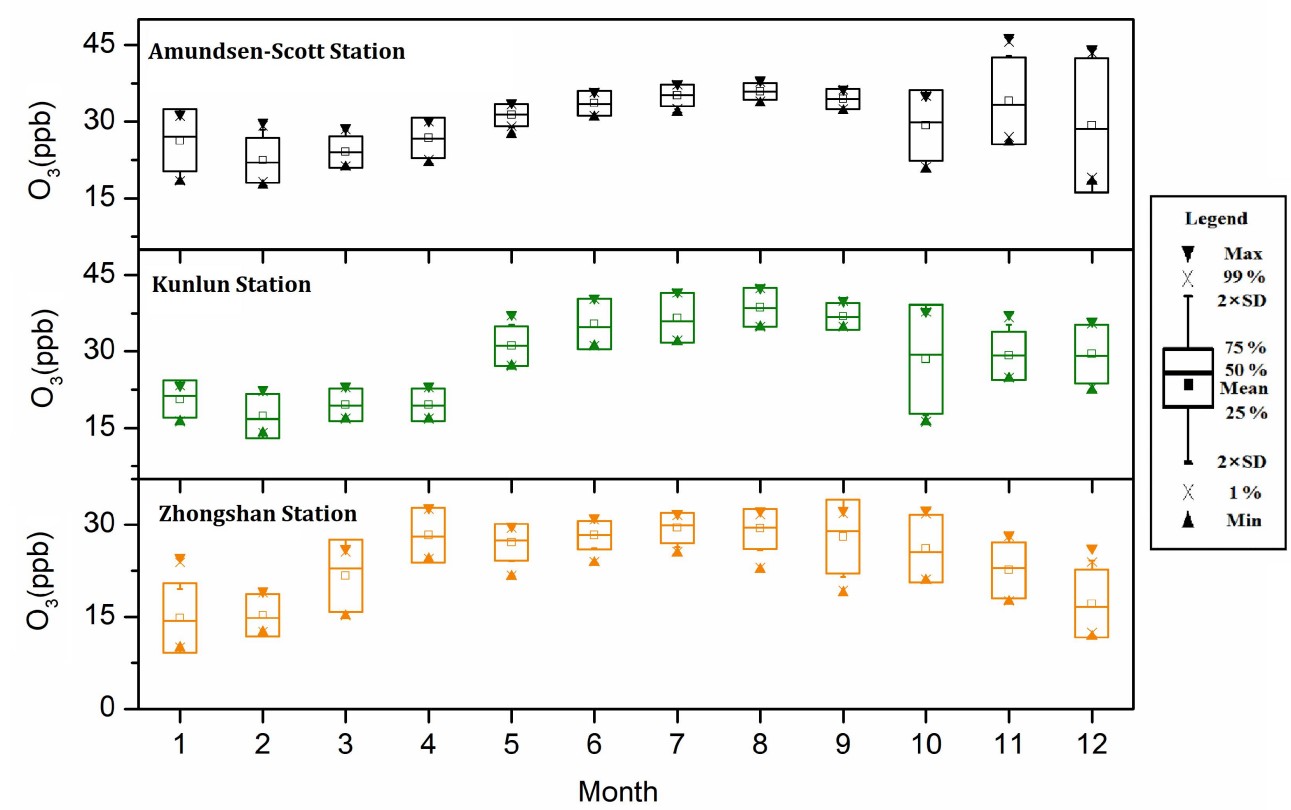

**Figure 3. Monthly average and statistical parameters of near-surface ozone at the Amundsen-Scott Station, Kunlun Station and Zhongshan Station during 2016**





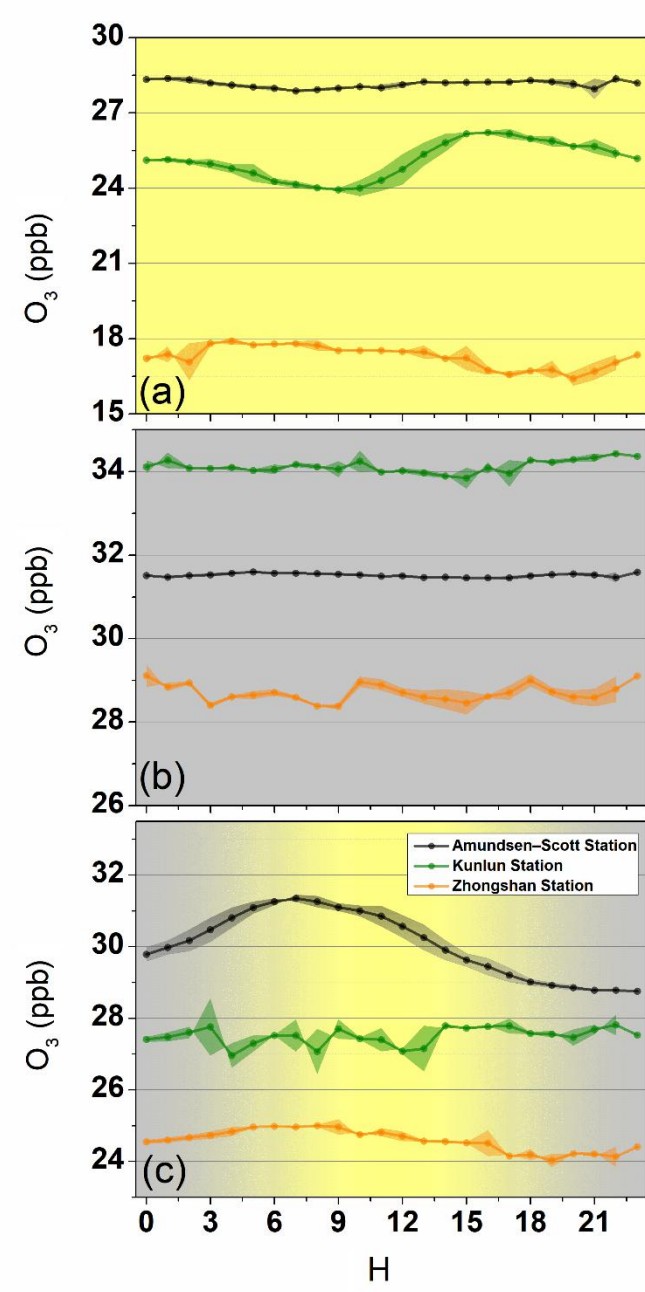

**Figure 4. Mean diurnal variations in near-surface ozone concentrations during polar day (a), polar night (b) and on a normal day (c), respectively, at the Amundsen-Scott Station, Kunlun Station and Zhongshan Station.**





**Figure 5. (a, b and c) The OEEs and (d, e and f) averaged distribution of OEE occurrence among the different months of 2016 at the three stations.** $Monthly\ frequency = \frac{times\ of\ OEE\ day\ of\ the\ month}{All\ days\ of\ the\ month}$; $Annual\ frequency = \frac{times\ of\ OEE\ day\ of\ the\ year}{All\ days\ of\ the\ year}$.






**Figure 6. Above ground level (m) of back trajectories for OEEs during polar day (a) and polar night (b); conditional probability field of the PSCF for the occurrence of OEEs at DA (c); pressure height (hPa) of back trajectories during polar day OEEs (d) and polar night OEEs (e).**


**Figure 7. (a) Ozone variations at Dome A and the South Pole; (b) the variability of ozone mixing ratios over the vertical profile; (c) the variability of potential vorticity over the vertical profile; (c) the variability of temperature over the vertical profile; (e) time series (daily 00:00, 6:00, 12:00, 18:00) of near-surface ozone, STT events (550 hPa, PVU>2) and 4 m wind speed (hourly average). All the results were calculated for polar night at DA.**


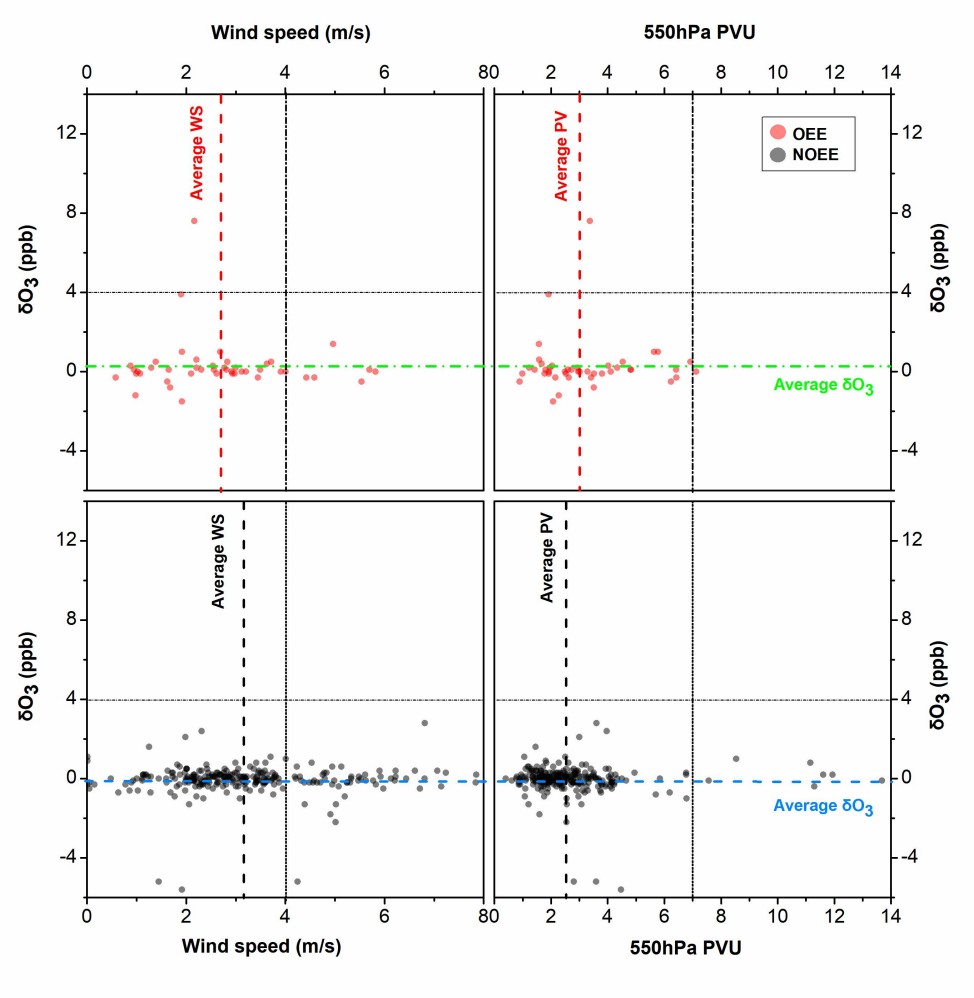

**Figure 8. Wind speed, 550 hPa potential vorticity and $\delta O_3$ statistical distribution around OEEs (red dots) and NOEEs (grey dots) at DA during polar night. Here, $\delta O_3$ represents the growth rate of near-surface $O_3$ concentration,**
**calculated by equation:** $\delta O_3 = \dfrac{\text{The O3 concentration at } T_n - \text{The O3 concentration at } T_{n-1}}{\text{Time difference of } T_n \text{ and } T_{n-1}}$.



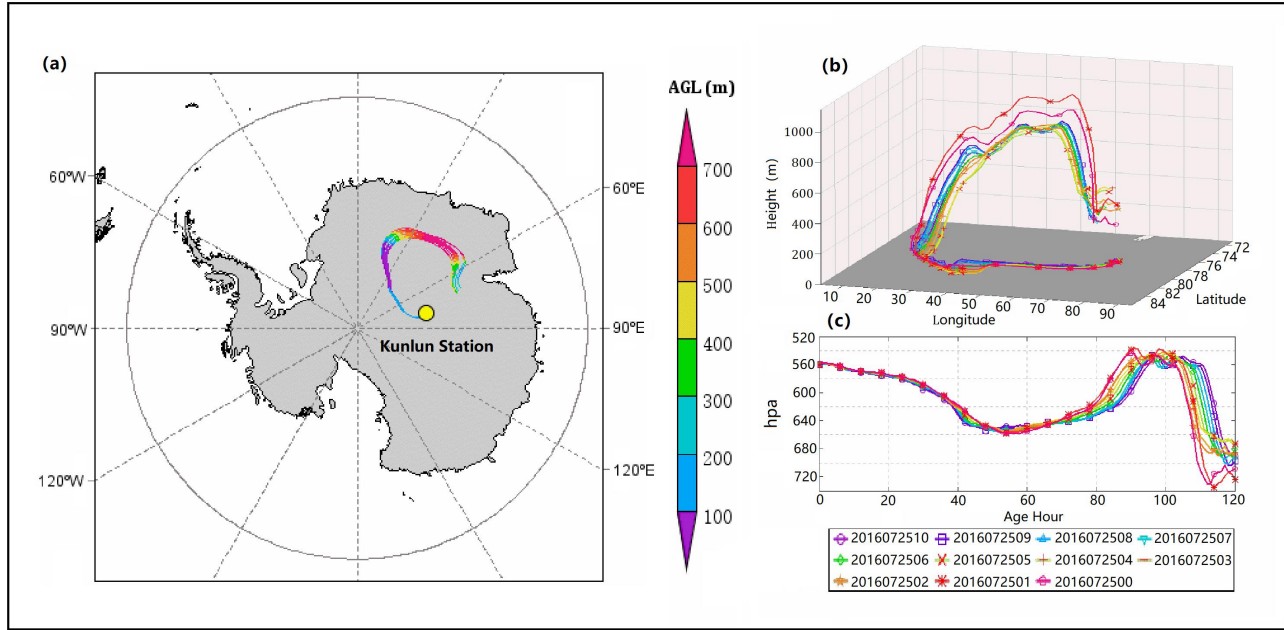

**Figure 9. (a) 5-day backward HYSPLIT trajectories from 25 June 00:00 to 25 June 10:00 for each measurement hour; (b) 3-D view of backward trajectory heights (m above ground level), longitude and latitude; (c) pressure height (hPa)**

**of the backward trajectories.**







**Figure 10. (a)** The potential vorticity at 550 hPa, **(b)** ozone concentration at 550 hPa, **(c)** U component of wind at 550 hpa and **(d)** Relative humidity at 550 hpa of Antarctica from 21st to 29th July 2016. The red dot represents the Kunlun Station.







**Table 1. The occurrence probability statistics of OEEs (A1), NOEEs (A0), STT events (B1), and NSTT events (B0).**

$$Frequency = \frac{\text{times of Event } (A*, B*)}{\text{total number of times}} \, .$$

| Events | Frequency of both phenomena occur |
|---|---|
| (A0,B0) | 12.42% |
| (A0,B1) | 38.49% |
| (A1,B0) | 16.29% |
| (A1,B1) | 32.79% |
