# Peer review of "Year-round record of near-surface ozone and "O3 enhancement events" (OEEs) at Dome A, East Antarctica"

_Atmospheric Chemistry and Physics, 2019_

## Short Comment (SC1) · 22 Jan 2020

I am glad that the authors expressed interest in the STEFLUX tool for analyzing the stratosphere-to-troposphere exchange at Dome A. However, I would like to ask the authors some clarifications on the use of the tool that they state in the paper.

In particular, what do the authors mean with Lines 339–343? The tropopause within STEFLUX is defined as the "dynamical tropopause", i.e., the combination of the $\pm2$ pvu potential vorticity isosurfaces and the 380 K isentrope, as also stated in Škerlak et al. (2014). Furthermore, STEFLUX involves the declaration of a "target box" of interest, which is taken as a reference 3D area for calculating the STT events. I am

quite confused because I do not see any definition of this "target box"; instead, the authors indicate the combination of PVU>2 and 300 hPa for analyzing the transport.

Last but not least, as indicated in the STEFLUX reference paper (Putero et al., 2016), unfortunately the code is not publicly available yet: the outputs of the tool are available on request by writing an email and by specifying the "target box" characteristics and the period of study chosen. I would be more than happy to collaborate with the authors, but so far none of the STEFLUX paper authors has received any request with regard to this.

Putero, D., Cristofanelli, P., Sprenger, M., Škerlak, B., Tositti, L., and Bonasoni, P.: STE-FLUX, a tool for investigating stratospheric intrusions: application to two WMO/GAW global stations, Atmos. Chem. Phys., 16, 14203–14217, https://doi.org/10.5194/acp-16-14203-2016, 2016.
Škerlak, B., Sprenger, M., and Wernli, H.: A global climatology of stratosphere–troposphere exchange using the ERA-Interim data set from 1979 to 2011, Atmos. Chem. Phys., 14, 913–937, https://doi.org/10.5194/acp-14-913-2014, 2014.
* * *

---

## Author Comment (AC1) · 5 Feb 2020

Thanks for your comments. As your said, the code of STEFLUX tool is not publicly available. When analyzing the data, we have asked for your help through Researchgate two months ago. So we turned to do analysis according the STEFLUX discipline, which defined the "dynamical tropopause" (ie, the combination of the $\pm 2$ pvu potential vorticity isosurfaces and the 380 K isentrope, as also stated in Škerlak et al. (2014)). Furthermore, STEFLUX involves the declaration of a "target box "of interest, which is taken as a reference 3D area for calculating the STT events. In this article, we replace the 3D area to a vertical profile above Dome A. As all the trajectories should

be converge above Dome A, the situation that the potential vorticity of 550hpa above 2 pvu or less than -2 pvu can be determined as once STT. But this method is not as good as STEFLUX, which can better screen out potential vorticity anomalies on trajectories in the 3D area, so the screened STT is incomplete. However, it is certain that the potential vorticity anomalies above Dome A at 550hpa are indicative. Besides of that, It is really glad that this study can draw your attention.We hope to do further analysis if you can help on STEFLUX tool. Thank you again for your kind suggestions.

---

## Referee Comment (RC1) · Anonymous Referee #2 · 6 Feb 2020

Reviewer's comments on the paper by Ding et al. entitled "Year-round record of near-surface ozone and "O3 enhancement events" (OEEs) at Dome A, East Antarctica" submitted to Atmospheric Chemistry and Physics

The manuscript is within the scope of ACP. It presents scientifically significant material based on surface ozone measurements at three Antarctic stations. Of especial importance are data of measurements at Dome Argus, the highest Antarctic plateau ($\sim$4000 m above sea level). However I have a lot of comments to the manuscript, which are listed below. The manuscript needs major revision.

General comments

1. One significant disadvantage of the manuscript is that some explanations of analysis results look like mere assertions. They are specified in more detail in the specific comments section.

2. The authors repeatedly expressed about importance of photochemical source of near-surface ozone in the Antarctic without providing evidence of it. Presumably they do not have clear idea of photochemical production of tropospheric ozone. See especially page 10.

3. Inconsistent scientific language is often used in the manuscript. English should be generally improved.

4. The potential source contribution function (PSCF) is corrected by multiplying it by some weights suggested earlier by other authors. However these weights are arbitrary and do not have any physical or mathematical reason. They modify arbitrary the distribution of the PSCF but do not allow estimating its statistical significance. I suppose that analysis of the PSCF distribution has to be done with accounting for statistical significance. Estimating statistical significance should take into account the fact that close-in-time trajectories are not independent. Without knowing whether the PSCF distribution is statistically significant one cannot rely on Fig. 6. Perhaps the following paper will help: Shikurov and Shukurova, Source regions of ammonium nitrate, ammonium sulfate, and natural silicates in the surface aerosols of Moscow oblast, Izvestiya, Atmos. Oceanic Phys. 2017, v. 53, p. 316-325, doi: 10.1134/S0001433817030136.

5. Potential vorticity (PV) in the southern hemisphere polar stratosphere is generally negative. However values of PV in Fig. 7 are of inverse sign. This contradicts also to PV distribution in Fig. 10.

6. Values of ozone concentration are given with excessive accuracy. One decimal place is enough.

7. There are no references to Fig. 4 and Fig. 7e in the text.

[Figure]

8. Some works that are referenced to in the text are absent in the reference list.

Specific comments

L45-46. Add reference(s) to confirm this.

L61-62. Add reference(s) regarding the depth of the mixing layer.

L73-76. The downward transport of ozone is important not only on high-altitude terrains. Note also that stratospheric ozone in the polar regions can be transported to the troposphere not only during intrusion events but also as a result of slow but prolonged subsidence. In this sentence, references would be more appropriate to papers concerning polar regions (e.g., Gruzdev and Sitnov 1993; Roscoe 2004, Possible descent across the "Tropopause" in Antarctic winter, Adv. Space Res., v. 33, p. 1048-1052; Greenslade et al. 2017, Stratospheric ozone intrusion events and their impacts on tropospheric ozone in the Southern Hemisphere, Atmos. Chem. Phys. V. 17, p. 10269–10290).

L91-92. Unclear. Why does it lead to overestimation?

L125-126. Specify address.

L154. What is PM?

L180-184. This paragraph is somewhat misleading. It reduces the ozone annual variation to change between polar day and polar night. However Fig. 2 shows that large values of ozone concentration peculiar to polar night are also observed for long time intervals before or/and after the polar night period. Similarly, low ozone concentration values peculiar to polar day are observed after the polar day period.

L185. Wrong statement. According to Fig. 2, Ozone concentration at the SP during polar night is generally less than at the Kunlun station.

L191-192. Gruzdev et al. –> Gruzdev and Sitnov. Oltmans et al. 1976 and Ghude et al. 2010 are absent in the reference list. Probably you mean Oltmans and Komhyr

1976, Surface ozone in Antarctica, JGR, v. 81, p. 5359-5364.

L193-196. Unfounded statements. Please confirm these by references or give clear arguments.

L198-199. Unreasonable explanation. Why weaker variability is due to short polar night?

L200-201. This explanation is not sufficiently reasoned since it refers to literature sources one of which is absent in the reference list and the other is an abstract.

L204-205. This explanation is not sufficiently reasoned since it does not follow from the references given.

L205-206. Misconception. Enhanced variability does not require a special ozone source.

L207-209. The explanation is unfounded.

L231-232. Papers by Oltmans et al. 1976 are absent in the reference list (see comment to L191-192). Gruzdev et al. is also absent in the reference list. However it is relevant and can be added: Gruzdev, Elokhov, Makarov and Mokhov, 1993, Some recent results of Russian measurements of surface ozone in Antarctica. A meteorological interpretation, Tellus, v. 45B, p. 99-105.

L219-234. It would be relevant to refer to Fig. 4 here. One interesting feature in Fig. 4 is the presence of a specific and very regular diurnal variation at the DA station during the polar day period. You could try to associate it with the slope katabatic winds which have diurnal cycle in summer (see aforementioned reference to Gruzdev et al. 1993). Although these winds are most prominent off the plateau they, due to their large horizontal scale, can induce slow subsidence of the air in the boundary layer over plateau and therefore influence the surface ozone concentration because of vertical ozone gradient.

L241-249. This part should be revised or removed.

L241-242. Are hydrocarbons really produced in surface snow?

L243. Wrong reaction.

L245. What do you mean by a chain reaction?

L245-246. Inconsistency: production occurs when loss (destruction) occurs.

L248-249. Why the cold is the reason of the variation?

L259. What is meant by a well-mixed state? Does it have to do with atmospheric mixing?

L258-265. This procedure is not completely clear and internally contradictory. First, it is hypothesized that data falling out of the Gaussian distribution are "abnormal". But then the Gaussian fit is applied to these data.

L267 and further. Two significant digits are enough.

L285. Do you mean the time or spatial scale?

L296. Air mass circulation? What is it? In meteorology, air mass is a volume of air which covers many hundreds or thousands kilometers in horizontal direction and hundreds meters or a few kilometers in vertical direction.

L297-309. See general comment 4. It is very probable that at least part of the PSCF is statistically insignificant. From my point of view, the main conclusion from the back trajectory analysis is that all the 5-day trajectories depicted in Figs 6a, b are located around the plateau and do not have their origin out of the continent.

L310. Simulated? Did you do your own simulations or use HYSPLIT?

L315. Jones et al. 1999 is absent in the reference list.

L317-318. A very probable reason is that the DA station is higher and therefore closer

to the tropopause.

L319. Do you mean the stratospheric polar vortex? Why do you mention it here? How is it related to ground level ozone?

L354-355. This explanation is unclear.

L359-369. This analysis is vague due to many reasons, see below.

L362-364. Bad language.

L263. September is not presented in Fig. 7.

L363-364. On what basis do you conclude about "extensive turbulence". The only source of turbulence in polar night is dynamical instability. But according to your data mentioned on page 15 the wind velocity was small during OEE events.

L363-365. I do not agree with this conclusion. Analysis of Figs 7a and c shows that there is no good correspondence between ozone maxima at Fig. 7a and subsidence of potential vorticity in Fig. 7c.

L365-366. The 50-200 hPa layer is not presented in Fig. 7.

L367. On what basis do you conclude that turbulence near the tropopause affects directly ozone? Do you really believe that there is intensive turbulence near the tropopause which is defined as a most statically stable layer?

L374. Which two events? The corresponding number is absent in the table.

L376-377. It is obvious, by definition of OEE, that increase during OEE is larger.

L380. What is PBLs? And what do lower PBLs mean?

Section4.3.3. Do not confuse vorticity with vortex.

L402-403. Negative value cannot be larger than positive value.

Technical corrections

L16. from –> at

L16. Specify that the Zhongshan Station is coastal.

L28. "account for" is probably wrong word.

L100. monitored –> measured

L104. spatial temporal –> spatial and temporal

L115. Give here the full name of the station.

L118. transported –> transferred

L123. related coefficients –> appropriate correlation coefficients

L178. experienced –> is characterized by

L192. stable –> less changeable

L193. variable –> more changeable

L309. pressure altitude?

L332. What is SI?

L336. stratosphere intrusion –> stratospheric intrusion

L337. stratospheric-affected –> stratosphere-affected

L340. define –> determine

L357. transmission –> transport

Figure 6c. The color scale used does not allow distinguishing peculiarities of the PSCF distribution.

---

## Referee Comment (RC2) · Anonymous Referee #3 · 8 Feb 2020

This paper is within the scope of ACP and presents a potentially interesting observational record by investigating an interesting scientific topic. However, the used methodologies are not adequate (in some cases - see the use of PV - wrong) and, also for these reasons, the conclusions are far to be robust and mostly based on qualitative and arbitrary interpretation of data. Moreover, the manuscript suffers of strong deficiencies in the vocabulary and in the quality of figures. Finally, I did not see any acknowledgments to people or Institutions providing the ozone data from South Pole station: from which data repository has been this dataset obtained?

Thus, I'm sorry but I have to recommend rejection or a complete re-submission of a

new manuscript when the following shortcoming will be fixed.

SPECIFIC COMMENTS:

Abstract: line 25: "To explain this unique finding, the occurrence of stratospheric intrusion (stratosphere-to-troposphere, STT) events was studied with the Stratosphere-to-Troposphere Exchange Flux (STEFLUX) tool". Here the author claimed STEFLUX was used in this work: unfortunately this is not the case (see also the related comment by one of the STEFLUX's authors). STEFLUX is a code developed by Putero et al (2016), see https://www.atmos-chem-phys.net/16/14203/2016/acp-16-14203-2016.pdf. No indication of the real use of STEFLUX can be found along the paper. At some point the authors claimed they selected back-trajectories coming from region with PV > 2 pvu (by the way: in the southern polar stratosphere the PV is negative, so this is wrong!): this is a rather simple filtering of back-trajectories far to be the application (or a replication) of STELFLUX. Thus, this sentence should be removed from the abstract.

Line 58 -72 (page 4): this section is almost a "cut-and-paste" from a paper by other authors (Cristofanelli et al., Analysis of multi-year near-surface ozone observations at the WMO/GAW "Concordia" station).

Section 2.1: the experiments is not well described. As an instance no information are provided about the set-up of the measurement system as well as used materials. No information about the application of a Quality Assurance strategy and good/standard practices. No reference to the adoption of (international) calibration scale. No details about the execution of the intercomparison with the travelling standard Thermo 49i-PS: the linear correlation coefficient is not sufficient to assess the overall quality of measurement (what about the total uncertainty)?

Line 113: vocabulary issue: "Model-49iPS UV absorptive ozone calibrator" should be "UV-absorption ozone calibrator Thermo 49i-PS"

Line 128: why data filtering? In general, I feel dangerous to automatically eliminate

data without motivation (i.e. error codes in the internal diagnostic, extremely inconsistent values,. . .).

Section 2.2 Meteorological simulations should be renamed as "Air mass back-trajectory calculations"

Section 2.2: a very poor description of the methodology (and strategy) for back-trajectory calculation is provided (no indication about time resolution of back-trajectories and frequency of their calculation). A discussion about usability and limitation of the use of these kind of back-trajectories based on coarse meteorological data is missing (e..g. it seems that the authors did not perform any sensitivity study to evaluate the impact of selecting different altitude or geographical position of the trajectory arrival point, which is a rather common practice to evaluate associated uncertainty). The authors mentioned that a clustering has been performed but they not provide any information about the clustering methodology nor provided evidences for cluster calculation in the paper. Vocabulary issue: "and the back-up time was 120 h". What is the back-up time?

Line 170: the assigned weights look arbitrary. No explanation or motivation provided.

Line 185: this sentence is not clear at all

Line 194: "In Antarctica, the emissions of ozone precursors are generally less than those at mid and low latitudes". Which precursors are emitted in Antarctica? By which process? What do you mean with "generic less"? Please try to be quantitative.

Line 207: "Specifically, the largest standard deviation was observed in October at DA because of multiple influences, including photochemical reactions by ozone precursors and ultraviolet radiation, photolysis reactions by strengthened ultraviolet radiation, and external air masses from the coast." These are only assumptions: not proofs are provided by the authors

Line 216: vocabulary issue: "Is there ozone exchange happening?" Ozone is trans-

ported not "exchanged".

Section 3: Overall, I think that the analysis of diurnal variation is not well executed in Section 3. What "normal days" are? If the authors' goal was to investigate the diurnal variability of ozone some relative measure should be used instead of actual mixing ratio (see for instance the earlier work by Helmig et al., 2007: "A review of surface ozone in the polar regions").

Line 225: "Because of the limited number of normal days, the diurnal concentration fluctuated". If the dashed area represents a confidence interval (not explained in the figure), the diel (not "diurnal": vocabulary issue) cycle looks well consistent and not "erratic", instead. On the contrary, it is the green series that looks more "noisy". Please avoid using this kind of background colors in the Figure 4 plots. Is time expressed as UTC or what else in Figure 4? The diel variability of ozone (even when evident, see green line in plot 4a or black line in plot 4c) is not explained or motivated enough by the authors.

Line 230: I do not agree. This can suggest that local photochemistry cannot have a role. But probably, if you consider the transport time, the integrated contribution of photochemistry related with snowpack NOx emissions can be relevant. This should be better assessed in the paper.

Line 241 – 245: This part is confuse and the description of cycle leading to ozone production is not correct. Sorry but I cannot really understand why the cold environment can motivate the daytime variability at NA.

Line 254 – 264: again, this is mostly a cut-and-past from an already published paper.

Figure 6: The analysis and interpretation of back-trajectory analysis presented in Figure 6 is not robust at all. Firstly the conditional probability should be calculated for winter and summer, if you want to demonstrate a prominent role of STT versus other processes in winter. From Figure 6, it looks that only a small number of TRJ are used

for this analysis (how much?): unfortunately this strongly limits the statistical robustness of results (that, in any case, do not support STT occurrence). Moreover, I'm not able to see any difference between back-trajectories in polar night/day that you used for motivate the role of STT during the winter.

Line 339: "Here, we use STEFLUX to 340 identify STT events and define the height of tropospheric potential vorticity PVU = 2." You did not use STEFLUX, actually. Moreover, in the Southern Hemisphere medium-high latitude, stratospheric air-masses can be traced setting PV < - 2 pvu and not PV > 2 pvu!

Line 370: "To quantitatively analyse the influence of STT events on OEEs, we examined 370 the appearance of STT events above DA and found that STT events (550 hPa PV>2 PVU) accounted for 55% of the polar night in 2016." This is wrong: firstly, to trace stratospheric air-masses, you should detect PV values lower than – 2 pvu. Secondly, as clearly see by Figure 7 at 550 hPa the PV variability is affected by non-adiabatic process occurring near the surface and thus it cannot be used to trace STE.

From Figure 7 is not possible to see any obvious correlation between ozone at DA and the downward transport of stratospheric air masses: the supposed link between high near-surface O3 at DA and occurrence of STE is not supported by a quantitative analysis (only a qualitative comment to Figure 7 is provided). Moreover, the wrong detection methodology used to identify the STE events brings an evident overestimation of STT occurrence: all the winter period (except August) appeared to be affected by STT (even without effect on near-surface O3, see e.g. the period from 6/20 to 7/15 which not support your hypothesis).

Finally, Figure 8 does not provide any reasonable support to the hypothesis that STE are driving O3 variability during winter. I do not see any evident differences between OEE and NOEE. It is not clear why using the rate of change of the hourly O3 m.r. instead of the actual O3 m.r.

---

## Referee Comment (RC3) · Anonymous Referee #1 · 9 Mar 2020

I agree with most of the comments and criticism that has been voiced already by the two other reviewers. This manuscript presents surface ozone data from a badly under-sampled region on Earth. This makes me wish to eventually see this work published. However, the experimental description, and the data presentation and interpretation, as well as the writing of the manuscript need substantial additional work before it meets my expectation for peer-reviewed publication. Below are some specific comments in addition to the points already raised by the two other reviewers.

Line 20: Clarify which data go with which station. Reduce significant figures of the averaged results here and in remainder of the text. Explain what the error margins are

(e.g. 1-sigma variability of hourly data?).

Line 47: Define NOx when it's mentioned for the first time.

Line 64: There are further publications that should be considered in the discussion of ozone chemistry in Antarctica: [Bauguitte et al., 2011; Davis et al., 2004; Davis et al., 2008; Helmig et al., 2008a; Helmig et al., 2008b; Neff et al., 2008; Oltmans et al., 2008]

Line 111: The 2B ozone monitor does not quite reach the analytical accuracy and precision of regular benchtop ozone analyzers. It can also have quite some sensitivity drifts over time. The performance of the analyzers should be presented in more detail. Calibration results/graphs should be provided as Supplemental Material.

Line 117: Please provide more information on the reference ozone monitor. This is not a commonly recognized instrument.

Line 125: Give credit, possibly offer co-authorship to the P.I.s and agencies that produced the data from South Pole.

Line 131: What caused the loss of data coverage at Kunlun Station?

Line 155: Define 'PM'.

Line 176: 'Concentration' is the wrong term as ozone data are presented as molar ratio, not as concentration.

Line 179: Specify if you mean average, median, or ? higher mole fractions.

Line 182: Define the polar day and night windows by day of year margins.

Line 189: I found this whole section hard to read and comprehend.

Line 219: Same here. What do you actually mean by 'diurnal variability'?

Line 224: How can you state that the diurnal concentration fluctuated greatly if the standard deviation is just 0.7 ppb?

[Figure]

Line 244: There is a rich body of more recent ozone snow photochemistry literature that should be considered in this discussion as well.

Line 422: The Neff et al., 2008, paper is about South Pole.

Line 430-432: I have been wondering about this all along reading this manuscript. Is this SST discussion even worth the effort given these pretty obvious limitations?

Line 436-437: Absolutely not acceptable. Quality-controlled final data should be made available in a well-recognized public data archive.

Line 636: I would prefer defining the time windows by day of year. There is no diurnal radiation or temperature cycle at South Pole. How do the authors explain the diurnal behavior seen in the ozone 'on a normal day'?

Line 639: Explain abbreviations in the graphs in the figure legend.

Line 640: I would expect the annual frequency results to be much smaller than the monthly results. How can they be this similar?

Line 643: Trajectory colors are hard to differentiate.

Line 649: Harmonize time stamps between figures.

Line 656: What are you trying to show with this figure? I don't really see a convincing dependency?

References

Bauguitte, S. J. B., N. Brough, M. M. Frey, A. E. Jones, D. J. Maxfield, H. K. Roscoe, M. C. Rose, and E. W. Wolff (2011), A network of autonomous surface ozone monitors in Antarctica: technical description and first results, Atmospheric Measurement Techniques, 4(4), 645-658, doi:10.5194/amt-4-645-2011.

Davis, D., G. Chen, M. Buhr, J. Crawford, D. Lenschow, B. Lefer, R. Shetter, F. Eisele, L. Mauldin, and A. Hogan (2004), South Pole NOx chemistry: an assessment of fac-

tors controlling variability and absolute levels, Atmospheric Environment, 38(32), 5375-5388.

Davis, D. D., et al. (2008), A reassessment of Antarctic plateau reactive nitrogen based on ANTCI 2003 airborne and ground based measurements, Atmospheric Environment, 42(12), 2831-2848.

Helmig, D., B. Johnson, S. J. Oltmans, W. Neff, F. Eisele, and D. D. Davis (2008a), Elevated ozone in the boundary layer at South Pole, Atmospheric Environment, 42(12), 2788-2803, doi:10.1016/j.atmosenv.2006.12.032.

Helmig, D., B. J. Johnson, M. Warshawsky, T. Morse, W. D. Neff, F. Eisele, and D. D. Davis (2008b), Nitric oxide in the boundary-layer at South Pole during the Antarctic Tropospheric Chemistry Investigation (ANTCI), Atmospheric Environment, 42(12), 2817-2830.

Neff, W., D. Helmig, A. Grachev, and D. Davis (2008), A study of boundary layer behavior associated with high NO concentrations at the South Pole using a minisodar, tethered balloon, and sonic anemometer, Atmospheric Environment, 42(12), 2762-2779.

Oltmans, S., B. J. Johnson, and D. Helmig (2008), Episodes of high surface-ozone amounts at South Pole during summer and their impact on the long-term surface-ozone variation, Atmospheric Environment, 42, 2804-2816.

———————————————————

---

## Author Comment (AC2) · 6 Apr 2020

**Reply for Anonymous Referee #1**

**I agree with most of the comments and criticism that has been voiced already by the two other reviewers. This manuscript presents surface ozone data from a badly undersampled region on Earth. This makes me wish to eventually see this work published. However, the experimental description, and the data presentation and interpretation, as well as the writing of the manuscript need substantial additional work before it meets my expectation for peer-reviewed publication. Below are some specific comments in addition to the points already raised by the two other reviewers.**

**Total response:** Thank you for your comments and suggestions on our article. We revised and explained the errors and insufficient explanations in the article one by one according to the suggestions. The article has undergone great changes: (1) we have made great amendments to the contents and illustrations of section 3, adding references and deleting subjective judgments. (2) We have greatly revised the content of Section 4.2. We have re-analyzed this section by re-determining the weight of PSCF. (3) We made a major revision to section 4.3, contacted Dr. Putero, and cooperated in STEFLUX analysis. The conclusion of the article has been corrected accordingly.

Thank you for your constructive and insightful criticism and advice. We addressed all the points raised by the reviewer as summarized below.

**Line 20: Clarify which data go with which station. Reduce significant figures of the averaged results here and in remainder of the text. Explain what the error margins are (e.g. 1-sigma variability of hourly data?).**

Reply: The summary has been modified accordingly. The error range here refers to the standard deviation of the annual hourly average concentration.

*"The annual mean values at the three stations (DA, SP and ZS) were 29.2 $\pm$ 7.5 ppb, 29.9 $\pm$ 5.0 ppb and 24.1 $\pm$ 5.8 ppb. "*

**Line 47: Define NOx when it's mentioned for the first time.**

Reply: Thanks for your suggestion. We have modified the original text and supplemented the definition of NOx.

*"Ozone ($O_3$) photochemical production in the troposphere occurs by hydroxyl radical oxidation of carbon monoxide (CO), methane (CH4) and non-methane hydrocarbons (generally referred to as NMHC) in the presence of **nitrogen oxides (NOx )** (Monks et al., 2015)"*

**Line 64: There are further publications that should be considered in the discussion of ozone chemistry in Antarctica: [Bauguitte et al., 2011; Davis et al., 2004; Davis et al.,**

**2008; Helmig et al., 2008a; Helmig et al., 2008b; Neff et al., 2008; Oltmans et al., 2008]**

Reply: Thanks for your suggestion, this error has been corrected. We added references and reinterpreted the contents.

*These processes are capable of driving the seasonality of near-surface O3 over the Antarctic Plateau (e.g., Crawford et al., 2001; Legrand et al., 2009), thus potentially providing a significant input of O3 to the whole Antarctic region (e.g., Legrand et al., 2016; **Bauguitte et al., 2011**).Indeed, **Helmig et al. (2008a,b)** provide further insight into the vigorous photochemistry and ozone production that result from the highly elevated levels of nitrogen oxides (NOx) in the Antarctic surface layer. During stable atmospheric conditions (which typically existed during low wind and fair sky conditions) ozone accumulated in the surface layer to reach up to twice its background concentration. **Neff et al. (2008a)** provide the earlier results that shallow mixing layers associated with light winds and strong surface stability can be among the dominant factors leading to high NO levels were repeated. As shown in Cristofanelli et al. (2008) and Legrand et al. (2016), due to air mass transport, the photochemically produced O3 in the PBL over the Antarctic Plateau can affect the O3 variability thousands of km away from the emission area.*

**Line 111: The 2B ozone monitor does not quite reach the analytical accuracy and precision of regular benchtop ozone analyzers. It can also have quite some sensitivity drifts over time. The performance of the analyzers should be presented in more detail. Calibration results/graphs should be provided as Supplemental Material.**

Reply: According to your suggestion, this part has been modified and replaced by new analysis and description. We found that there was drift in the instrument, but the data was available after cross concentration calibration. We make a detailed supplement to the performance characteristics of the instrument and the calibration chart.

*The Model 205 Dual Beam Ozone Monitor makes use of two detection cells to improve precision, baseline stability, and response time. In the Dual Beam instrument, UV light intensity measurements I0(ozone-scrubbed air) and I(unscrubbed air) are made simultaneously. Combined with other improvements, this instrument is the fastest UV-based ozone monitor on the market, while have small size, weight, And power requirements characteristics. Fast measurements are particularly desirable for unattended station, aircraft and balloon measurements where high spatial resolution is desired. The Model 205 dual beam ozone monitor is an Environmental Protection Agency (EPA) Federal equivalent method (FEM). In order to better understand the monitoring accuracy and stability of model 205, we conducted a comparative test. During the experiment, the Thermo 49i produced by the Thermo Fisher Scientific company was used as a comparison with the Model 205 (Wang et al., 2017), and a UV-absorption ozone calibrator Thermo 49i-PS is used to generate the standard for Model 205 and Thermo 49i, During July 28,2016 to July 29, natural air was continuously monitored by the two instruments at the same time. We found that the accuracy of the Model 205 was similar with Thermo 49i, so it can be concluded that Model 205 has a good applicability in practical work. Because of the extreme environment, we can't do quarterly*

*calibration like the Global Atmosphere Watch (GAW) stations. we could only calibrate the instrument during austral summer with a UV-absorption ozone calibrator Thermo 49i-PS. However, the appropriate correlation coefficients (r) were all greater than 0.99 in Jan 2017. The drift of the instrument is within the allowable range (Supplementary Material-Fig. S1).*

[Figure]

**Figure S1. Dynamic calibration of the linear equation about Model 205**

**Line 117: Please provide more information on the reference ozone monitor. This is not a commonly recognized instrument.**

Reply: According to your suggestion, this part has been modified and replaced by new analysis and description. We make a detailed supplement to the performance characteristics of the instrument.

*The Model 205 Dual Beam Ozone Monitor makes use of two detection cells to improve precision, baseline stability, and response time. In the Dual Beam instrument, UV light intensity measurements I0(ozone-scrubbed air) and I(unscrubbed air) are made simultaneously. Combined with other improvements, this instrument is the fastest UV-based ozone monitor on the market, while have small size, weight, And power requirements characteristics. Fast measurements are particularly desirable for unattended station, aircraft and balloon measurements where high spatial resolution is desired. The Model 205 dual beam ozone monitor is an Environmental Protection Agency (EPA) Federal equivalent method (FEM). In order to better understand the monitoring accuracy and stability of model 205, we conducted a comparative test. During the experiment, the Thermo 49i produced by the Thermo Fisher Scientific company was used as a comparison with the Model 205 (Wang et al., 2017), and a UV-absorption ozone calibrator Thermo 49i-PS is used to generate the standard for Model 205 and Thermo 49i, During July 28,2016 to July 29, natural air was continuously monitored by the two instruments at the same time. We found that the accuracy of the Model 205 was similar with Thermo 49i, so it can be concluded that Model 205 has a good applicability in practical work. Because of the extreme environment, we can't do quarterly calibration like the Global Atmosphere Watch (GAW) stations. we could only calibrate the instrument during austral summer with a UV-absorption ozone calibrator Thermo 49i-PS.*

*However, the appropriate correlation coefficients (r) were all greater than 0.99 in Jan 2017. The drift of the instrument is within the allowable range (Supplementary Material-Fig. S1).*

**Line 125: Give credit, possibly offer co-authorship to the P.I.s and agencies that produced the data from South Pole.**

Reply: According to your suggestion, we have made additional changes to this issue.

*The Amundsen-Scott Station (89 ° 59'51.19 "S, 139 ° 16'22.41" E, altitude 2835 m) is located at the SP and operated by the United States. The near-surface ozone data were downloaded **from the Earth System Research Laboratory Global Monitoring Division under the NOAA (https://www.esrl.noaa.gov/gmd/dv/data).***

**Line 131: What caused the loss of data coverage at Kunlun Station?**

Reply: Due to the time when the Chinese Antarctic scientific expedition arrived at Dome A, the instrument made normal observation from January 20, 2016. As a result, January's data is missing a lot.

**Line 155: Define 'PM'.**

Reply: The words used here are not accurate. The original text is changed here to: "If the total number of end points that fall in the cell is nij and there are mij points for which the measured ozone parameter exceeds a criterion value selected for this parameter, then the conditional probability, the PSCF, can then be defined as:"

**Line 176: 'Concentration' is the wrong term as ozone data are presented as molar ratio, not as concentration.**

Reply: The error has been changed. Concentration -> molar ratio

**Line 179: Specify if you mean average, median, or ? higher mole fractions.**

Reply: According to your suggestion, this part has been modified and replaced by new analysis and description.

*The annual mean molar ratios of near-surface ozone at DA, the SP, and the ZS were 29.2± 7.5 ppb, 29.9 ± 5.0 ppb and 24.1 ± 5.8 ppb, respectively. **The maximum annual mean molar ratio** reached 42.5 ppb, 46.4 ppb and 32.8 ppb, and the **minimum annual mean molar ratios** were 14.0 ppb, 10.9 ppb and 9.9 ppb, respectively. **The inland stations are characterized by higher annual mean molar ratios** than the coastal station.*

**Line 182: Define the polar day and night windows by day of year margins.**

Reply: We have followed your suggestion and made some supplementary changes to this sentence.

*In Figure 3, we define the polar day and night windows by day of year margins and used different shading colours to signify the polar day and polar night.*

**Line 189: I found this whole section hard to read and comprehend.**

Reply: In order to explain the seasonal variation characteristics of the three sites more clearly. The section 3.2 has changed a lot.

*In this part, we determine Oct-Mar as the warm season and Apr-Sept as the cold season, similar to the definition of polar day and night.*

*In agreement with previous studies (Oltmans et al., 1976; Gruzdev et al., 1993; Ghude S D et al., 2005), the concentrations of near-surface ozone at the three stations were high and less changeable during the cold season and low and more changeable during the warm season (Figure 4).In Antarctica, the emissions of ozone precursors are generally less than those at mid and low latitudes, whereas ultraviolet radiation is relatively strong; thus, When there is solar radiation,the depletion effect is much greater than the effects from photochemical reactions during the warm season (Schnell et al., 1991). As previous studies have explained, during the polar night, due to the lack of light, photochemical reaction stopped. And Due to the lack of loss effect, the ozone concentration gradually increases and the fluctuation becomes smaller. During the polar night, the monthly variation of surface ozone in ZS is smaller, higher than SP, but lower than DA. However, due to strong UV radiation in low latitude areas and the presence of bromine controlled ozone depletion events in coastal areas, the station shows a large seasonal variation during the non polar night (Wang et al., 2011; Prados-Roman et al., 2017). However, at the Amundsen-Scott Station, the largest standard deviation was observed in December, similar to the characteristics at DC from November to December (Legrand et al., 2009; Cristofanelli et al., 2018). Figures 3 and 4 show that the near-surface ozone showed obviously larger variations at DA than at the SP during the polar night.Due to the difference of geographical location, the meteorological conditions of DA and SP are different. The abnormal fluctuation of ozone concentration over DA during the polar night may be related to its special geographical environment.*

*As mentioned in the introduction section, mountainous topography/mountain waves may disturb advection transport in the stratosphere and lead to downward transportation to the troposphere (Robinson et al., 1983). DA is the summit of the east Antarctic Ice Sheet, and the tropospheric depth is only ~4.6 km (Liang et al., 2015), which favours exchange between the stratosphere and troposphere. However, the topography in this area is very flat and creates a disadvantage for mountain waves. Is there ozone transport happening? We will analyse and discuss this question in section 4.*

**Line 219: Same here. What do you actually mean by 'diurnal variability'?**

Reply: It is the change characteristics of the diurnal variation of each month.

**Line 224: How can you state that the diurnal concentration fluctuated greatly if the standard deviation is just 0.7 ppb?**

Reply: Yes, the expression here is ambiguous. We have reinterpreted and charted Section 3.3. We found that the diurnal variation fluctuation of three stations was less than 1 ppb.

**Line 244: There is a rich body of more recent ozone snow photochemistry literature that should be considered in this discussion as well.**

Reply: Due to the lack of measured NOx data, we can only illustrate the average daily characteristics of ground ozone concentration at three stations according to the chart. However, the general characteristics show that the daily photochemistry reactions did not have the main impacts on the overall characteristics of near-surface ozone at the three stations.

For the difference of diurnal variation of three stations in different months, more measured data should be used for accurate analysis in the future.

**Line 422: The Neff et al., 2008, paper is about South Pole.**

Reply: We have revised it in the article.

**Line 430-432: I have been wondering about this all along reading this manuscript. Is this SST discussion even worth the effort given these pretty obvious limitations?**

Reply: The section 4.2 has changed a lot. Indeed, STEFLUX was applied, state here the importance of investigating STT events in Antarctica, which is a region poorly studied in this respect, and where large differences between the estimates of STT frequencies exist.

**Line 436-437: Absolutely not acceptable. Quality-controlled final data should be made available in a well-recognized public data archive.**

Reply: We will make available in a well-recognized public data archive.

**Line 636: I would prefer defining the time windows by day of year. There is no diurnal radiation or temperature cycle at South Pole. How do the authors explain the diurnal behavior seen in the ozone 'on a normal day'?**

Reply:According to your suggestion, this part has been modified and replaced by new analysis and description. I agree with you. There is no diurnal radiation or temperature cycle at three stations. The section 3.3 has changed a lot. The error has been changed.

*To characterize the typical monthly O3 diurnal variations at these three stations, We analyzed the mean diurnal variations of three stations (Figure 4) and the standard deviation of mean diurnal variations (Figure 5). At DA, the mean diurnal concentrations of each month were relatively steady, with the standard deviation of the mean diurnal concentration of each month is lower than 0.4ppb. At the SP, the mean diurnal concentrations of each month were less changeable too. Except for December, the standard deviation of mean diurnal concentration was lower than 0.3ppb. At the ZS, except for October, the standard deviation of mean diurnal concentration in other months is greater than that in other two stations. In particular, the standard deviation of the mean diurnal concentration of ZS in September, November and December exceeded 0.5ppb. The mean diurnal variations in different time periods were not obvious, and the mean diurnal concentrations of the three stations fluctuated within a range of less than 1 ppb, indicating that daily photochemistry reactions did not have the main impacts on the overall characteristics of near-surface ozone at the three stations. The magnitude of the diurnal variation was low, which is similar to the variations found at other Antarctic stations (Gruzdev et al., 1993; Ghude S D et al., 2005; Oltmans et al., 2008).*

[Figure]

**Figure 4. Mean diurnal variations in near-surface ozone concentrations at the Amundsen-Scott Station (a), Kunlun Station (b) and Zhongshan Station (c) during 2016**

[Figure]

**Figure 5. The Standard deviation of mean diurnal variations in near-surface ozone concentrations at the Amundsen-Scott Station, Kunlun Station and Zhongshan Station during 2016**

**Line 639: Explain abbreviations in the graphs in the figure legend.**

Reply: The error has been changed.

**Line 640: I would expect the annual frequency results to be much smaller than the monthly results. How can they be this similar?**

Reply: The Reviewer is right, but this is related to the error in the formula within the caption.

$$Monthly\ frequency = \frac{number\ of\ OEE\ days\ for\ each\ month}{number\ of\ days\ in\ the\ month}$$

$$Annual\ frequency = \frac{number\ of\ OEE\ days\ for\ each\ month}{total\ number\ of\ OEE\ days}.$$

**Line 643: Trajectory colors are hard to differentiate.**

Reply: The error has been changed.

**Line 649: Harmonize time stamps between figures.**

Reply: The error has been changed.

**Line 656: What are you trying to show with this figure? I don't really see a convincing dependency?**

Reply: According to the content of the article, the figure has been modified and corrected.

**References**

**Bauguitte, S. J. B., N. Brough, M. M. Frey, A. E. Jones, D. J. Maxfield, H. K. Roscoe, M. C. Rose, and E. W. Wolff (2011), A network of autonomous surface ozone monitors in Antarctica: technical description and first results, Atmospheric Measurement Techniques, 4(4), 645-658, doi:10.5194/amt-4-645-2011.**

**Davis, D., G. Chen, M. Buhr, J. Crawford, D. Lenschow, B. Lefer, R. Shetter, F. Eisele, L. Mauldin, and A. Hogan (2004), South Pole NOx chemistry: an assessment of fac-**

---

## Author Comment (AC3) · 6 Apr 2020

**Reply for Anonymous Referee #2**

**Reviewer's comments on the paper by Ding et al. entitled "Year-round record of nearsurface ozone and "O₃ enhancement events" (OEEs) at Dome A, East Antarctica" submitted to Atmospheric Chemistry and Physics**

**The manuscript is within the scope of ACP. It presents scientifically significant material based on surface ozone measurements at three Antarctic stations. Of especial importance are data of measurements at Dome Argus, the highest Antarctic plateau (∽ 4000 m above sea level). However I have a lot of comments to the manuscript, which are listed below. The manuscript needs major revision.**

Total response: Thank you for your comments and suggestions on our article. We revised and explained the errors and insufficient explanations in the article one by one according to the suggestions. The article has undergone great changes: (1) we have made great amendments to the contents and illustrations of section 3, adding references and deleting subjective judgments. (2) We have greatly revised the content of Section 4.2. We have re-analyzed this section by re-determining the weight of PSCF. (3) We made a major revision to section 4.3, contacted Dr. Putero, and cooperated in STEFLUX analysis. The conclusion of the article has been corrected accordingly.

Thank you for your constructive and insightful criticism and advice. We addressed all the points raised by the reviewer as summarized below.

**1. One significant disadvantage of the manuscript is that some explanations of analysis results look like mere assertions. They are specified in more detail in the specific comments section.**

Reply: The new manuscript has been checked carefully and we think these assertions has been improved.

**2. The authors repeatedly expressed about importance of photochemical source of near-surface ozone in the Antarctic without providing evidence of it. Presumably they do not have clear idea of photochemical production of tropospheric ozone. See especially page 10.**

Reply:According to your suggestion, this part has been modified and replaced by new analysis and description. Please find in the Line 255~269.

**3. Inconsistent scientific language is often used in the manuscript. English should be generally improved.**

Reply: The full text has been retouched and edited, and the English language has been

improved.

**4. The potential source contribution function (PSCF) is corrected by multiplying it by some weights suggested earlier by other authors. However these weights are arbitrary and do not have any physical or mathematical reason. They modify arbitrary the distribution of the PSCF but do not allow estimating its statistical significance. I suppose that analysis of the PSCF distribution has to be done with accounting for statistical significance. Estimating statistical significance should take into account the fact that close-in-time trajectories are not independent. Without knowing whether the PSCF distribution is statistically significant one cannot rely on Fig. 6. Perhaps the following paper will help: Shikurov and Shukurova, Source regions of ammonium nitrate, ammonium sulfate, and natural silicates in the surface aerosols of Moscow oblast, Izvestiya, Atmos. Oceanic Phys. 2017, v. 53, p. 316-325, doi: 10.1134/S0001433817030136.**

Reply: According to your suggestion, we have performed new experiments to better clarify the PSCF. We first cluster all the air mass trajectories during OEE and NOEE. Then we recalculate the track weights of different periods. Through recalculation of PSCF, we find that there are obvious differences in the potential pollution sources in different periods. The results are in good agreement with those of cluster analysis. The new weight calculation and graph are shown below.

$$
W_{ij(NOEE)} = \begin{cases} 1.00 & nij > 12\,Nave \\ 0.70 & 12\,Nave > nij > 3\,Nave \\ 0.42 & 3\,Nave > nij > 1.5\,Nave \\ 0.05 & Nave > nij \end{cases}
$$

$$
W_{ij(OEE)} = \begin{cases} 1.00 & nij > 8\,Nave \\ 0.70 & 8\,Nave > nij > 2\,Nave \\ 0.42 & 2\,Nave > nij > 1\,Nave \\ 0.05 & Nave > nij \end{cases}
$$

[Figure]

Figure 7. Likely source areas of surface ozone at Kunlun Station during the NOEE (a) and OEE (b) identified using PSCF (Potential Source Contribution Function).

[Figure]

Figure 8. Backward HYSPLIT trajectories for each measurement day (gray lines in the Fig.8a), and mean back trajectory for 3 HYSPLIT clusters (colored lines in the Fig.8a, 3D view shown in the Fig.8b) arriving at Kunlun Station during the NOEE. Mean trajectory of 3 HYSPLIT clusters arriving at Kunlun Station during the NOEE. Subplot (c) shows the range of surface ozone mixing ratios measured at Kunlun Station by cluster.

[Figure]

Figure 9. Same as Fig. 8, but for OEEs.

**5. Potential vorticity (PV) in the southern hemisphere polar stratosphere is generally negative. However values of PV in Fig. 7 are of inverse sign. This contradicts also to PV distribution in Fig. 10.**

Reply: We have made great changes to the content of the fourth section of the article, and used the STEFLUX tool to analyze STT events. In combination with your suggestion, we have corrected the error.

**6. Values of ozone concentration are given with excessive accuracy. One decimal place is enough.**

Reply:According to your suggestion. The ozone concentration value in this paper is accurate to one decimal place.

**7. There are no references to Fig. 4 and Fig. 7e in the text.**

Reply:According to your suggestion, this part has been modified and replaced by new analysis and description. please find in the Line 257 and Line286.

**8.Some works that are referenced to in the text are absent in the reference list.**

**Specific comments**

**L45-46. Add reference(s) to confirm this.**

Reply: The original text is changed to: ozone ($O_3$) photochemical production in the troposphere occurs by hydroxyl radical oxidation of carbon monoxides (CO), metal ($CH_4$) and non metal hydrocarbons (generally referred to as NMHC) in the presence of nitrogen oxides (NOx) (Monks et al., 2015).

**L61-62. Add reference(s) regarding the depth of the mixing layer.**

Reply: According to your suggestion, we have added reference(s) regarding the depth of the mixing layer. Please find in the Line 67.

Berman S., Ku J Y., Rao S T.: Spatial and Temporal Variation in the Mixing Depth over the Northeastern United States during the Summer of 1995, Journal of Applied Meteorology, 38(12):1661-1673, https://doi.org/10.1175/1520-0450(1999)038<1661:SATVIT>2.0.CO;2, 1999.

**L73-76. The downward transport of ozone is important not only on high-altitude terrains. Note also that stratospheric ozone in the polar regions can be transported to the troposphere not only during intrusion events but also as a result of slow but prolonged subsidence. In this sentence, references would be more appropriate to papers concerning polar regions (e.g., Gruzdev and Sitnov 1993; Roscoe 2004, Possible descent across the "Tropopause" in Antarctic winter, Adv. Space Res., v. 33, p. 1048- 1052; Greenslade et al. 2017, Stratospheric ozone intrusion events and their impacts on tropospheric ozone in the Southern Hemisphere, Atmos. Chem. Phys. V. 17, p. 10269-10290).**

Reply: According to your suggestion, I have made changes and added explanations. The original text is changed to: "The near surface ozone concentrations at high-elevation sites can also be increased by the downward transport of ozone rich air from the stratosphere during deep conversion and stratosphere to troposphere transport (STT) events. More, the stratospheric ozone in the polar regions can be transported to the troposphere not only during intrusion events but also as a result of slow but prolonged subsidence. (e.g., Gruzdev et al., 1993; Roscoe et al., 2004; Greenslade et al. 2017,)."

**L91-92. Unclear. Why does it lead to overestimation?**

Reply: The words used here are not accurate. We changed "overestimation" to "inaccurate estimation".Please find in the Line 104.

**L125-126. Specify address.**

Reply: The Amundsen-Scott Station (89° 59'51.19 "S, 139° 16'22.41" E, altitude 2835 m) is located at the SP and operated by the United States. The near-surface ozone data were downloaded from the Earth System Research Laboratory Global Monitoring Division under the NOAA (https://www.esrl.noaa.gov/gmd/dv/data).

**L154. What is PM?**

Reply: The words used here are not accurate. The original text is changed here to: "If the total number of end points that fall in the cell is nij and there are mij points for which the measured ozone parameter exceeds a criterion value selected for this parameter, then the conditional probability, the PSCF, can then be defined as:"

**L180-184. This paragraph is somewhat misleading. It reduces the ozone annual variation to change between polar day and polar night. However Fig. 2 shows that large values of ozone concentration peculiar to polar night are also observed for long time intervals before or/and after the polar night period. Similarly, low ozone concentration values peculiar to polar day are observed after the polar day period.**

Reply: Yes, the comparison of the concentration characteristics of polar day and night is mainly in Figure 3.

**L185. Wrong statement. According to Fig. 2, Ozone concentration at the SP during polar night is generally less than at the Kunlun station.**

Reply: Yes, there is a mistake here.The SP had the highest near-surface ozone concentration during non-polar night. The average concentration during this period was 28.1 ppb.

**L191-192. Gruzdev et al. -> Gruzdev and Sitnov. Oltmans et al. 1976 and Ghude**

**et al. 2010 are absent in the reference list. Probably you mean Oltmans and Komhyr**

**1976, Surface ozone in Antarctica, JGR, v. 81, p. 5359-5364.**

Reply:There is a mistake in citation. Thank you for your correction. The revised version has been amended.

*Oltmans, S. J Komhyr, W. D.: Surface ozone in Antarctica, Journal of Geophysical Research, v. 81, p. 5359-5364. http://dx.doi.org/10.1029/jc081i030p05359,1976.*

**L193-196. Unfounded statements. Please confirm these by references or give clear arguments.**

Reply:In Antarctica, the emissions of ozone precursors are generally less than those at mid and low latitudes, whereas ultraviolet radiation is relatively strong; thus, When there is solar radiation,the depletion effect is much greater than the effects from photochemical reactions

during the warm season (Schnell et al., 1991).

**L198-199. Unreasonable explanation. Why weaker variability is due to short polar night?**

Reply: During the polar night, due to the lack of light, photochemical reaction stopped. And Due to the lack of loss effect, the ozone concentration gradually increases and the fluctuation becomes smaller. During the polar night, the monthly variation of surface ozone in ZS is smaller, higher than SP, but lower than DA.

**L200-201. This explanation is not sufficiently reasoned since it refers to literature sources one of which is absent in the reference list and the other is an abstract.**

Reply: However, due to strong UV radiation in low latitude areas and the presence of bromine controlled ozone depletion events in coastal areas, the station shows a large seasonal variation during the non polar night (Wang et al., 2011; Prados-Roman et al., 2017).

**L204-205. This explanation is not sufficiently reasoned since it does not follow from the references given.**

Reply: This part is reinterpreted. The original part has been deleted.

However, at the Amundsen-Scott Station, the largest standard deviation was observed in December, similar to the characteristics at DC from November to December (Legrand et al., 2009; Cristofanelli et al., 2018). Figures 3 and 4 show that the near-surface ozone showed obviously larger variations at DA than at the SP during the polar night.Due to the difference of geographical location, the meteorological conditions of DA and SP are different. The abnormal fluctuation of ozone concentration over DA during the polar night may be related to its special geographical environment.

**L205-206. Misconception. Enhanced variability does not require a special ozone source.**

Reply: The original part has been deleted.

*However, at the Amundsen-Scott Station, the largest standard deviation was observed in December, similar to the characteristics at DC from November to December (Legrand et al., 2009; Cristofanelli et al., 2018). Figures 2 and 3 show that the near-surface ozone showed obviously larger variations at DA than at the SP during the polar night.Due to the difference of geographical location, the meteorological conditions of DA and SP are different. The abnormal fluctuation of ozone concentration over DA during the polar night may be related*

*to its special geographical environment.*

**L207-209. The explanation is unfounded.**

Reply: The original part has been deleted.

**L231-232. Papers by Oltmans et al. 1976 are absent in the reference list (see comment to L191-192). Gruzdev et al. is also absent in the reference list. However it is relevant and can be added: Gruzdev, Elokhov, Makarov and Mokhov, 1993, Some recent results of Russian measurements of surface ozone in Antarctica. A meteorological interpretation, Tellus, v. 45B, p. 99-105.**

Reply: Thanks for your suggestion, this error has been corrected. And added references and explanations.

**L219-234. It would be relevant to refer to Fig. 4 here. One interesting feature in Fig. 4 is the presence of a specific and very regular diurnal variation at the DA station during the polar day period. You could try to associate it with the slope katabatic winds which have diurnal cycle in summer (see aforementioned reference to Gruzdev et al. 1993). Although these winds are most prominent off the plateau they, due to their large horizontal scale, can induce slow subsidence of the air in the boundary layer over plateau and therefore influence the surface ozone concentration because of vertical ozone gradient.**

Reply: Thanks for your suggestion. The section 3.3 has changed a lot. The error has been changed.

*To characterize the typical monthly O3 diurnal variations at these three stations, we analysed the mean diurnal variations of O3 at the three stations (Figure 4) and the standard deviation of the mean diurnal variations (Figure 5). At the DA site, the mean diurnal concentrations of each month were relatively steady, and with the standard deviation of the mean diurnal concentration of each month was lower than 0.4 ppb. At the SP, the mean diurnal concentrations were less variable as well. Except for December, the standard deviation of the mean diurnal concentration was lower than 0.3 ppb. At the ZS, except for October, the standard deviation of the mean diurnal concentration is greater than that in the other two stations. In particular, the standard deviation of the mean diurnal concentration of the ZS in September, November and December exceeded 0.5 ppb. The mean diurnal variations in different time periods were not obvious, and the mean diurnal concentrations of the three stations fluctuated within a range of less than 1 ppb, indicating that daily photochemistry reactions did not have the main impacts on the overall characteristics of near-surface ozone at the three stations. The magnitude of the diurnal variation was low, which is similar to the*

variations found at other Antarctic stations (Gruzdev et al., 1993; Ghude et al., 2005; Oltmans et al., 2008).

[Figure]

**Figure 4. Mean diurnal variations in near-surface ozone concentrations at the Amundsen-Scott Station (a), Kunlun Station (b) and Zhongshan Station (c) during 2016**

[Figure]

**Figure 5. The Standard deviation of mean diurnal variations in near-surface ozone concentrations at the Amundsen-Scott Station, Kunlun Station and Zhongshan Station during 2016**

**L241-249. This part should be revised or removed.**

Reply: Thanks for your suggestion. The section 3.3 has changed a lot. This part has been removed.

**L241-242. Are hydrocarbons really produced in surface snow?**

Reply: There is a mistake in the expression here. The photodegradation on the snow surface only releases NOx and does not contain hydrocarbons. The section 3.3 has changed a lot. This part has been removed.

**L243. Wrong reaction.**

Reply: Thanks for your suggestion, this error has been corrected. The section 3.3 has changed a lot.

**L245. What do you mean by a chain reaction?**

Reply: The section 3.3 has changed a lot. The error has been changed.

**L245-246. Inconsistency: production occurs when loss (destruction) occurs.**

Reply: The section 3.3 has changed a lot. The error has been changed.

**L248-249. Why the cold is the reason of the variation?**

Reply: The section 3.3 has changed a lot. The error has been changed.

**L259. What is meant by a well-mixed state? Does it have to do with atmospheric mixing?**

Reply: Here is a hypothesis, if the Gaussian distribution is consistent, it represents an idealized good mixing state of the atmosphere.

**L258-265. This procedure is not completely clear and internally contradictory. First, it is hypothesized that data falling out of the Gaussian distribution are "abnormal". But then the Gaussian fit is applied to these data.**

Reply: Thanks for your suggestion, this error has been corrected. We have reinterpreted and modified this part.

*The algorithm proposed in Cristofanelli et al. The analysis of OEEs was restricted to years 2016. Our method to select the days characterized by OEEs is based on a two-steps procedure. The first step consists in fitting the annual cycle of O3 mean daily values with a sinusoidal curve. This represents an "undisturbed" O3 annual cycle, not affected by the occurrence of summer O3 events. In the second step, the probability density function (PDF) of the deviations from the sinusoidal fit is computed, considering all of the daily data. Then, a Gaussian fit was applied to the obtained PDF. As reported by Giostra et al. (2011), the Gaussian distribution corresponds to a well-mixed state, given the hypothesis that instrumental errors and natural background variability follow a Gaussian distribution. The*

*deviations from the Gaussian distribution (calculated by the Origin 9© statistical tool) would indicate observations affected by non-background variability. To obtain a threshold value for selecting non-background O3 daily values possibly affected by "anomalous" O3 enhancements, we calculated a further Gaussian fit for the PDF points falling above 1 $\sigma$ (standard deviation) of the Gaussian PDF, and we considered the intersection between the two curves as our threshold value (i.e., 3.4 ppb at the SP, 3.4 ppb at DA, 2.5 ppb at the ZS).Figure 6a, 6b, 6c shows the OEEs and "NO O3 enhancement events" (NOEEs) at these three stations, while Figure 6d, 6e, 6f reports the distribution frequency of OEE.*

**L267 and further. Two significant digits are enough.**

Reply: Some of them are reduced.

**L285. Do you mean the time or spatial scale?**

Reply: In this paper, we consider the spatial scale.

**L296. Air mass circulation? What is it? In meteorology, air mass is a volume of air which covers many hundreds or thousands kilometers in horizontal direction and hundreds meters or a few kilometers in vertical direction.**

Reply: The error has been changed. Air mass circulation ->Air mass transport

**L297-309. See general comment 4. It is very probable that at least part of the PSCF is statistically insignificant. From my point of view, the main conclusion from the back trajectory analysis is that all the 5-day trajectories depicted in Figs 6a, b are located around the plateau and do not have their origin out of the continent.**

Reply: Thanks for your suggestion. In the new section 4.2, we analyze the potential sources of the backward trajectory during NOEE and OEE respectively. In this way, we can clearly see the difference between the potential sources of two different periods.

**L310. Simulated? Did you do your own simulations or use HYSPLIT?**

Reply: The section 4.2 has changed a lot. The error has been changed. We used HYSPLIT to cluster and analyze the potential sources of NOEEs and OEEs.

**L315. Jones et al. 1999 is absent in the reference list.**

Reply: The section 4.2 has changed a lot. The error has been changed.

**L317-318. A very probable reason is that the DA station is higher and therefore closer to the tropopause.**

Reply: The section 4.2 has changed a lot. The error has been changed.

**L319. Do you mean the stratospheric polar vortex? Why do you mention it here? How is it related to ground level ozone?**

Reply: The section 4.2 has changed a lot. The error has been changed.

**L354-355. This explanation is unclear.**

Reply: The fourth section has changed a lot. The original error no longer exists.

**L359-369. This analysis is vague due to many reasons, see below.**

Reply: The fourth section has changed a lot. The original error no longer exists.

**L362-364. Bad language.**

Reply: The fourth section has changed a lot. The original error no longer exists.

**L263. September is not presented in Fig. 7.**

Reply: The fourth section has changed a lot. The original error no longer exists.

**L363-364. On what basis do you conclude about "extensive turbulence". The only source of turbulence in polar night is dynamical instability. But according to your data mentioned on page 15 the wind velocity was small during OEE events.**

Reply: The fourth section has changed a lot. The original error no longer exists.

**L363-365. I do not agree with this conclusion. Analysis of Figs 7a and c shows that there is no good correspondence between ozone maxima at Fig. 7a and subsidence of potential vorticity in Fig. 7c.**

Reply: The fourth section has changed a lot. The original error no longer exists.

**L365-366. The 50-200 hPa layer is not presented in Fig. 7.**

Reply: The fourth section has changed a lot. The original error no longer exists.

**L367. On what basis do you conclude that turbulence near the tropopause affects directly ozone? Do you really believe that there is intensive turbulence near the tropopause which is defined as a most statically stable layer?**

Reply: The fourth section has changed a lot. The original error no longer exists.

**L374. Which two events? The corresponding number is absent in the table.**

Reply: The fourth section has changed a lot. The original error no longer exists.

**L376-377. It is obvious, by definition of OEE, that increase during OEE is larger.**

Reply: The fourth section has changed a lot. The original error no longer exists.

**L380. What is PBLs? And what do lower PBLs mean?**

**Section4.3.3. Do not confuse vorticity with vortex.**

Reply: The fourth section has changed a lot. The original error no longer exists.

**L402-403. Negative value cannot be larger than positive value.**

**Technical corrections**

Reply: The fourth section has changed a lot. The original error no longer exists.

**L16. from -> at**

Reply: The error has been changed.

**L16. Specify that the Zhongshan Station is coastal**.

Reply: Zhongshan Station ->Zhongshan Station (Southeast coast of Prydz Bay)

**L28. "account for" is probably wrong word.**

Reply: account for-> expound

**L100. monitored -> measured**

Reply: The error has been changed.

**L104. spatial temporal -> spatial and temporal**

Reply: The error has been changed.

**L115. Give here the full name of the station.**

Reply: The error has been changed.

**L118. transported -> transferred**

Reply: The error has been changed.

**L123. related coefficients -> appropriate correlation coefficients**

Reply: The error has been changed.

**L178. experienced -> is characterized by**

Reply: The error has been changed.

**L192. stable -> less changeable**

Reply: The error has been changed.

**L193. variable -> more changeable**

Reply: The error has been changed.

**L309. pressure altitude?**

Reply: The error has been changed. Pressure altitude corrected to pressure height.

**L332. What is SI?**

Reply: The words are wrong here. SI corrected to STT.

**L336. stratosphere intrusion -> stratospheric intrusion**

Reply: The error has been changed

**L337. stratospheric-affected -> stratosphere-affected**

Reply: The error has been changed

**L340. define -> determine**

Reply: The error has been changed.

**L357. transmission -> transport**

Reply: The error has been changed.

---

## Author Comment (AC5) · 6 Apr 2020

**Reply for Anonymous Referee #3**

**This paper is within the scope of ACP and presents a potentially interesting observational record by investigating an interesting scientific topic. However, the used methodologies are not adequate (in some cases - see the use of PV - wrong) and, also for these reasons, the conclusions are far to be robust and mostly based on qualitative and arbitrary interpretation of data. Moreover, the manuscript suffers of strong deficiencies in the vocabulary and in the quality of figures. Finally, I did not see any acknowledgments to people or Institutions providing the ozone data from South Pole station: from which data repository has been this dataset obtained?**

**Thus, I'm sorry but I have to recommend rejection or a complete re-submission of a**

**new manuscript when the following shortcoming will be fixed. SPECIFICCOMMENTS:**

**Abstract: line 25: "To explain this unique finding, the occurrence of stratospheric intrusion (stratosphere-to-troposphere, STT) events was studied with the Stratosphere-to- Troposphere Exchange Flux (STEFLUX) tool". Here the author claimed STEFLUX was used in this work: unfortunately this is not the case (see also the related comment by one of the STEFLUX's authors). STEFLUX is a code developed by Putero et al (2016), see https://www.atmos-chem-phys.net/16/14203/2016/acp-16-14203-2016.pdf. No indication of the real use of STEFLUX can be found along the paper. At some point the authors claimed they selected back-trajectories coming from region with PV>2 pvu (by the way: in the southern polar stratosphere the PV is negative, so this is wrong!): this is a rather simple filtering of back-trajectories far to be the application (or a replication) of STELFLUX. Thus, this sentence should be removed from the abstract.**

**Total response:** Thank you for your comments and suggestions on our article. We have made additional changes to this issue.

*The Amundsen-Scott Station (89°59'51.19"S, 139°16'22.41" E, altitude 2835 m) is located at the SP and operated by the United States. The near-surface ozone data were downloaded* **from the Earth System Research Laboratory Global Monitoring Division under the NOAA (https://www.esrl.noaa.gov/gmd/dv/data).**

We revised and explained the errors and insufficient explanations in the article one by one according to the suggestions. The article has undergone great changes: (1) we have made great amendments to the contents and illustrations of section 3, adding references and deleting subjective judgments. (2) We have greatly revised the content of Section 4.2. We have re-analyzed this section by re-determining the weight of PSCF. (3) We made a major revision to section 4.3, contacted Dr. Putero, and cooperated in STEFLUX analysis. The conclusion of the article has been corrected accordingly.

Thank you for your constructive and insightful criticism and advice. We addressed all the

points raised by the reviewer as summarized below.

**Line 58 -72 (page 4): this section is almost a "cut-and-paste" from a paper by other authors (Cristofanelli et al., Analysis of multi-year near-surface ozone observations at the WmO/GAW "Concordia" station).**

Reply: According to your suggestion, we have re edited this paragraph.

*Previous studies have shown that the near-surface O3 of Antarctica may be influenced by a number of climate related variables (Berman et al., 1999), i.e., the variation of UV flux caused by the variation of O3 column concentration over Antarctica (Jones and Wolff, 2003; Frey et al., 2015), accumulation and transport of long-distance, high concentration air masses (e.g., Legrand et al., 2016), and the depth of continental mixing layers. Many studies has observed summer episodes of "O3 enhancement events" (OEEs) in the Antarctic interior (e.g., Crawford et al., 2001; Legrand et al., 2009; Cristofanelli et al., 2018), and they attributed this phenomenon to the NOx emissions from snowpack and subsequent photochemical O3 production (for example, Jones et al., 2000). However, this may provide an input source for the entire Antarctic region (for example, Legrand et al., 2016; bauguitte et al., 2011). Indeed, Helmig et al. (2008a,b) provided further insight into the vigorous photochemistry and ozone production that result from the highly elevated levels of nitrogen oxides (NOx) in the Antarctic surface layer. During stable atmospheric conditions (which are typically observed during low wind and fair sky conditions) ozone accumulated in the surface layer to reach up to twice its background concentration. Neff et al. (2008a) show that shallow mixing layers associated with light winds and strong surface stability can be among the dominant factors leading to high NO levels. As shown in Cristofanelli et al. (2008) and Legrand et al. (2016), due to air mass transport, the photochemically-produced O3 in the PBL over the Antarctic Plateau can affect the O3 variability thousands of km away from the emission area.*

**Section 2.1: the experiments is not well described. As an instance no information are provided about the set-up of the measurement system as well as used materials. No information about the application of a Quality Assurance strategy and good/standard practices. No reference to the adoption of (international) calibration scale. No details about the execution of the intercomparison with the travelling standard Thermo 49iPS: the linear correlation coefficient is not sufficient to assess the overall quality of measurement (what about the total uncertainty)?**

Reply: According to your suggestion, we have made a supplementary explanation for the experimental part and supplemented the alignment chart of calibration, and the relevant data will also be submitted as required.

*The Kunlun Station (80°25'02"S, 77°06'59"E, altitude 4087 m) is located in the DA area, on the summit of the east Antarctic Ice Sheet (Figure 1). On the 1st of Jan 2016, we deployed*

*a Model 205 Dual Beam Ozone Monitor during the 33rd Chinese National Antarctic Research Expedition. The Model 205 Dual Beam Ozone Monitor makes use of two detection cells to improve precision, baseline stability, and response time. In the dual beam instrument, UV light intensity measurements I0 (ozone-scrubbed air) and I (unscrubbed air) are performed simultaneously. Combined with other improvements, this instrument is the fastest UV-based ozone monitor on the market, with such small size, weight, and power requirements characteristics. Fast measurements are particularly desirable for unattended stations, aircraft and balloon measurements, where high spatial resolution is desired. The Model 205 Dual Beam Ozone Monitor is an Environmental Protection Agency (EPA) federal equivalent method (FEM). To better understand the monitoring accuracy and stability of this device, we conducted a comparative test. During the experiment, the results of the Thermo 49i device produced by Thermo Fisher Scientific were compared to that of the Model 205 device(Wang et al., 2017), and a UV-absorption ozone calibrator Thermo 49i-PS was used to generate the standard for the Model 205 and Thermo 49i devices. From July 28 to July 29, natural air was continuously monitored by the two instruments at the same time. We found that the accuracy of the Model 205 device was similar with to that of the Thermo 49i device, so it can be concluded that Model 205 has a good applicability in practical work. Because of the extreme environment, we could not perform any quarterly calibrations, such as at Global Atmosphere Watch (GAW) stations. We could only calibrate the instrument during the austral summer with a UV-absorption ozone calibrator Thermo 49i-PS. However, the appropriate correlation coefficients (r) were all greater than 0.99 in January 2017, and the drift of the instrument was within the allowable range (Supplementary Material-Fig. S1).*

**Line 113: vocabulary issue: "Model-49iPS UV absorptive ozone calibrator" should be "UV-absorption ozone calibrator Thermo 49i-PS"**

Reply: According to your suggestion, we have made replacement and modification .

**Line 128: why data filtering? In general, I feel dangerous to automatically eliminate**

**data without motivation (i.e. error codes in the internal diagnostic, extremely inconsistent values,...).**

Reply: The data filtering here is mainly for Zhongshan station and Kunlun station. The deleted data includes: (1) missing measured value. (2) Monitoring data during calibration. (3) Severe fluctuation data in case of obvious pollution.

**Section 2.2 Meteorological simulations should be renamed as "Air mass back-trajectory calculations"**

Reply: According to your suggestion, we have made replacement and modification. Please find in the Line161

**Section 2.2: a very poor description of the methodology (and strategy) for back-trajectory calculation is provided (no indication about time resolution of back-trajectories and frequency of their calculation). A discussion about usability and limitation of the use of these kind of back-trajectories based on coarse meteorological data is missing (e..g. it seems that the authors did not perform any sensitivity study to evaluate the impact of selecting different altitude or geographical position of the trajectory arrival point, which is a rather common practice to evaluate associated uncertainty). The authors mentioned that a clustering has been performed but they not provide any information about the clustering methodology nor provided evidences for cluster calculation in the paper. Vocabulary issue: "and the back-up time was 120 h". What is the back-up time?**

Reply: According to your suggestion, we have made replacement and modificatio. Please find in the Line177~183.

*The integral error part of the trajectory calculation error can be estimated by simulating the backward trajectory at the end of the forward trajectory and comparing the differences of the tracks. The starting point of the backward integration is set as (77.12 ° E, 80.42 ° S, 20m AGL), the backward integration is 120h. And then the point reached at this time is taken as the starting point. At this moment as the initial time, the forward simulation is 120h. In this simulation experiment, the contribution of integration error to trajectory calculation error is very small Within 72 hours. With the extension of integration time, the integration error slightly increases.*

**Line 170: the assigned weights look arbitrary. No explanation or motivation provided.**

Reply:To expound the uncertainty due to the low values of nij, the PSCF values were scaled by a weighting function Wij (Polissar et al., 1999). We recalculated the weights of NOEE and OEE.

$$W_{ij(NOEE)} = \begin{cases} 1.00 & nij > 12Nave \\ 0.70 & 12Nave > nij > 3Nave \\ 0.42 & 3Nave > nij > 1.5Nave \\ 0.05 & Nave > nij \end{cases}$$

$$W_{ij(OEE)} = \begin{cases} 1.00 & nij > 8Nave \\ 0.70 & 8Nave > nij > 2Nave \\ 0.42 & 2Nave > nij > 1Nave \\ 0.05 & Nave > nij \end{cases}$$

[Figure]

Figure 7. Likely source areas of surface ozone at Kunlun Station during the NOEEs (a) and OEEs (b) identified using PSCF (Potential Source Contribution Function).

**Line 185: this sentence is not clear at all**

Reply:The sentence has been modified..Please find in the Line 225~227.

*Interestingly, the SP had the highest near-surface ozone concentration during non-polar night, whereas at DA presented the highest concentration occurred during polar night and the largest variation occurred at this site.*

**Line 194: "In Antarctica, the emissions of ozone precursors are generally less than those at mid and low latitudes". Which precursors are emitted in Antarctica? By which process? What do you mean with "generic less"? Please try to be quantitative.**

Reply: According to your suggestion, to explain the seasonal variation characteristics of the three sites more clearly, Section 3.2 has changed a lot.

*In this part, we define Oct-Mar as the warm season and Apr-Sept as the cold season, which is similarly to the definition of polar day and night.*

*In agreement with previous studies (Oltmans et al., 1976; Gruzdev et al., 1993; Ghude et al., 2005), the concentrations of near-surface ozone at the three stations were high and less variable during the cold season and low and more variable during the warm season (Figure 4). In Antarctica, the emissions of ozone precursors are generally less than those at mid and low latitudes, whereas ultraviolet radiation is relatively strong; thus, when solar radiation occurs, the depletion effect is much greater than the effects from photochemical reactions during the warm season (Schnell et al., 1991). As explained by previous studies, during the polar night, due to the lack of light, the photochemical reactions stoped. Moreover, due to the lack of loss effect, the ozone concentration gradually increased and the fluctuations became smaller. During the polar night, the monthly variation of surface ozone at ZS was lower than that at the DA but higher than that at the SP. However, due to strong UV radiation in the low*

*latitude areas and the presence of bromine controlled ozone depletion events in coastal areas, the ZS shows a large seasonal variations during the non-polar night (Wang et al., 2011; Prados-Roman et al., 2017). However, at the SP Station, the largest standard deviation was observed in December, similarly to the characteristics at Dome-C station (DC) from November to December (Legrand et al., 2009; Cristofanelli et al., 2018). Figures 3 and 4 show that the near-surface ozone showed obviously larger variations at the DA than the SP during the polar night, since, due to the different of geographical location, the meteorological conditions of DA and SP are different. The abnormal fluctuation of ozone concentration over the DA during the polar night may be related to its special geographical environment.*

*As mentioned in the introduction section, mountainous topography/mountain waves may disturb advection transport in the stratosphere and lead to downward transportation to the troposphere (Robinson et al., 1983). DA is on the summit of the east Antarctic Ice Sheet, and the tropospheric depth is only ~4.6 km (Liang et al., 2015), which favours exchange between the stratosphere and troposphere. However, the topography in this area is very flat and creates a disadvantage for mountain waves. Does ozone transport occur? We will analyse and discuss this question in section 4.*

**Line 207: "Specifically, the largest standard deviation was observed in October at DA because of multiple influences, including photochemical reactions by ozone precursors and ultraviolet radiation, photolysis reactions by strengthened ultraviolet radiation, and external air masses from the coast." These are only assumptions: not proofs are provided by the authors**

Reply: According to your suggestion, we have deleted this paragraph. In the last paragraph, we added references and explanations. Please find in the Line 231-248.

**Line 216: vocabulary issue: "Is there ozone exchange happening?" Ozone is transported not "exchanged".**

Reply: exchange -> transport

**Section 3: Overall, I think that the analysis of diurnal variation is not well executed in Section 3. What "normal days" are? If the authors' goal was to investigate the diurnal variability of ozone some relative measure should be used instead of actual mixing ratio (see for instance the earlier work by Helmig et al., 2007: "A review of surface ozone in the polar regions").**

Reply: Thanks for your advice. We recalculated the average daily variation of each month. The standard deviation of the diurnal variation of each month is calculated to express the characteristics of its diurnal fluctuation. There is no diurnal radiation or temperature cycle at three stations. The section 3.3 has changed a lot.

*To characterize the typical monthly O3 diurnal variations at these three stations, we analysed the mean diurnal variations of O3 at the three stations (Figure 4) and the standard deviation of the mean diurnal variations (Figure 5). At the DA site, the mean diurnal concentrations of each month were relatively steady, and with the standard deviation of the mean diurnal concentration of each month was lower than 0.4 ppb. At the SP, the mean diurnal concentrations were less variable as well. Except for December, the standard deviation of the mean diurnal concentration was lower than 0.3 ppb. At the ZS, except for October, the standard deviation of the mean diurnal concentration is greater than that in the other two stations. In particular, the standard deviation of the mean diurnal concentration of the ZS in September, November and December exceeded 0.5 ppb. The mean diurnal variations in different time periods were not obvious, and the mean diurnal concentrations of the three stations fluctuated within a range of less than 1 ppb, indicating that daily photochemistry reactions did not have the main impacts on the overall characteristics of near-surface ozone at the three stations. The magnitude of the diurnal variation was low, which is similar to the variations found at other Antarctic stations (Gruzdev et al., 1993; Ghude et al., 2005; Oltmans et al., 2008).*

**Line 225: "Because of the limited number of normal days, the diurnal concentration fluctuated". If the dashed area represents a confidence interval (not explained in the figure), the diel (not "diurnal": vocabulary issue) cycle looks well consistent and not "erratic", instead. On the contrary, it is the green series that looks more "noisy". Please avoid using this kind of background colors in the Figure 4 plots. Is time expressed as UTC or what else in Figure 4? The diel variability of ozone (even when evident, see green line in plot 4a or black line in plot 4c) is not explained or motivated enough by the authors.**

Reply: Thanks for your advice. We recalculated the average daily variation of each month. The standard deviation of the diurnal variation of each month is calculated to express the characteristics of its diurnal fluctuation.There is no diurnal radiation or temperature cycle at three stations. The section 3.3 has changed a lot. The error has been changed.

*To characterize the typical monthly O3 diurnal variations at these three stations, we analysed the mean diurnal variations of O3 at the three stations (Figure 4) and the standard deviation of the mean diurnal variations (Figure 5). At the DA site, the mean diurnal concentrations of each month were relatively steady, and with the standard deviation of the mean diurnal concentration of each month was lower than 0.4 ppb. At the SP, the mean diurnal concentrations were less variable as well. Except for December, the standard deviation of the mean diurnal concentration was lower than 0.3 ppb. At the ZS, except for October, the standard deviation of the mean diurnal concentration is greater than that in the other two stations. In particular, the standard deviation of the mean diurnal concentration of the ZS in September, November and December exceeded 0.5 ppb. The mean diurnal variations in different time periods were not obvious, and the mean diurnal concentrations of the three stations fluctuated within a range of less than 1 ppb, indicating that daily photochemistry reactions did not have the main impacts on the overall characteristics of near-surface ozone*

*at the three stations. The magnitude of the diurnal variation was low, which is similar to the variations found at other Antarctic stations (Gruzdev et al., 1993; Ghude et al., 2005; Oltmans et al., 2008).*

[Figure]

**Figure 4. Mean diurnal variations in near-surface ozone concentrations at the Amundsen-Scott Station (a), Kunlun Station (b) and Zhongshan Station (c) during 2016**

[Figure]

**Figure 5. The Standard deviation of mean diurnal variations in near-surface ozone concentrations at the Amundsen-Scott Station, Kunlun Station and Zhongshan Station during 2016**

**Line 230: I do not agree. This can suggest that local photochemistry cannot have a role. But probably, if you consider the transport time, the integrated contribution of photochemistry related with snowpack NOx emissions can be relevant. This should be better assessed in the paper.**

Reply: Due to the lack of measured NOx data, we can only illustrate the average daily characteristics of ground ozone concentration at three stations according to the chart. However, the general characteristics show that the daily photochemistry reactions did not have the main impacts on the overall characteristics of near-surface ozone at the three stations.

For the difference of diurnal variation of three stations in different months, more

measured data should be used for accurate analysis in the future.

**Line 241 一 245: This part is confuse and the description of cycle leading to ozone production is not correct. Sorry but I cannot really understand why the cold environment can motivate the daytime variability at NA.**

Reply: In the absence of measured evidence, I think your criticism is right. I deleted my subjective guesses, hoping to bring them to the next experiment.

**Line 254 一 264: again, this is mostly a cut-and-past from an already published paper.**

Reply: The sentence has been modified.

*Our method to select the days characterized by OEEs is based on the two-steps procedure shown in Cristofanelli et al. (2018). The first step consists in calculating the O3 annual cycle not affected by the OEEs (by using a sinusoidal fit), while the second one concerns the calculation of a probability density function (PDF) of the deviations from the sinusoidal fit, plus the application of a Gaussian fit to the obtained PDF. As reported in Giostra et al. (2011), the deviations from the Gaussian distribution (calculated by using the Origin 9© statistical tool) indicate observations affected by non-background variability.*

**Figure 6: The analysis and interpretation of back-trajectory analysis presented in Figure 6 is not robust at all. Firstly the conditional probability should be calculated for winter and summer, if you want to demonstrate a prominent role of STT versus other processes in winter. From Figure 6, it looks that only a small number of TRJ are used for this analysis (how much?): unfortunately this strongly limits the statistical robustness of results (that, in any case, do not support STT occurrence). Moreover, I'm not able to see any difference between back-trajectories in polar night/day that you used for motivate the role of STT during the winter.**

Reply: The fourth section has changed a lot. The original error no longer exists.

**Line 339: "Here, we use STEFLUX to 340 identify STT events and define the height of tropospheric potential vorticity PVU = 2." You did not use STEFLUX, actually. Moreover, in the Southern Hemisphere medium-high latitude, stratospheric air-masses can be traced setting PV < - 2 pvu and not PV > 2 pvu!**

Reply: The fourth section has changed a lot. The original error no longer exists.

**Line 370: "To quantitatively analyse the influence of STT events on OEEs, we examined 370 the appearance of STT events above DA and found that STT events (550 hPa PV>2 PVU) accounted for 55% of the polar night in 2016." This is wrong: firstly, to trace stratospheric air-masses, you should detect PV values lower than 一 2 pvu. Secondly, as clearly see by Figure 7 at 550 hPa the PV variability is affected by non-adiabatic process occurring near the surface and thus it cannot be used to trace STE.**

Reply: The fourth section has changed a lot. The original error no longer exists.

**From Figure 7 is not possible to see any obvious correlation between ozone at DA and the downward transport of stratospheric air masses: the supposed link between high near-surface O3 at DA and occurrence of STE is not supported by a quantitative analysis (only a qualitative comment to Figure 7 is provided). Moreover, the wrong detection methodology used to identify the STE events brings an evident overestimation of STT occurrence: all the winter period (except August) appeared to be affected by STT (even without effect on near-surface O3, see e.g. the period from 6/20 to 7/15 which not support your hypothesis).**

Reply: The fourth section has changed a lot. The original error no longer exists.

**Finally, Figure 8 does not provide any reasonable support to the hypothesis that STE are driving O3 variability during winter. I do not see any evident differences between OEE and NOEE. It is not clear why using the rate of change of the hourly O3 m.r. instead of the actual O3 m.r.**

Reply: The fourth section has changed a lot. The original error no longer exists.